# Pax6 limits the competence of developing cerebral cortical cells to respond to inductive intercellular signals

Martine Manuel[1☯], Kai Boon Tan[1☯], Zrinko Kozic[1], Michael Molinek[1], Tiago Sena Marcos[1], Maizatul Fazilah Abd Razak[1], Dániel Dobolyi[1¤], Ross Dobie[2], Beth E. P. Henderson[2], Neil C. Henderson[2], Wai Kit Chan[1], Michael I. Daw[1,3], John O. Mason[1], David J. Price[1]*

1 Simons Initiative for the Developing Brain, Patrick Wild Centre, University of Edinburgh, Edinburgh, United Kingdom, 2 Centre for Inflammation Research, University of Edinburgh, Queen's Medical Research Institute, Edinburgh, United Kingdom, 3 Zhejiang University-University of Edinburgh Institute, Zhejiang University, Haining, Zhejiang, People's Republic of China

☯ These authors contributed equally to this work.
¤ Current address: Wolfson Institute for Biomedical Research and Department of Neuroscience, Physiology and Pharmacology, University College London, London, United Kingdom
* David.Price@ed.ac.uk

**Data Availability Statement:** Data are either contained within the paper or are available at the European Nucleotide Archive (accession numbers:

## Abstract

The development of stable specialized cell types in multicellular organisms relies on mechanisms controlling inductive intercellular signals and the competence of cells to respond to such signals. In developing cerebral cortex, progenitors generate only glutamatergic excitatory neurons despite being exposed to signals with the potential to initiate the production of other neuronal types, suggesting that their competence is limited. Here, we tested the hypothesis that this limitation is due to their expression of transcription factor Pax6. We used bulk and single-cell RNAseq to show that conditional cortex-specific Pax6 deletion from the onset of cortical neurogenesis allowed some progenitors to generate abnormal lineages resembling those normally found outside the cortex. Analysis of selected gene expression showed that the changes occurred in specific spatiotemporal patterns. We then compared the responses of control and Pax6-deleted cortical cells to in vivo and in vitro manipulations of extracellular signals. We found that Pax6 loss increased cortical progenitors' competence to generate inappropriate lineages in response to extracellular factors normally present in developing cortex, including the morphogens Shh and Bmp4. Regional variation in the levels of these factors could explain spatiotemporal patterns of fate change following Pax6 deletion in vivo. We propose that Pax6's main role in developing cortical cells is to minimize the risk of their development being derailed by the potential side effects of morphogens engaged contemporaneously in other essential functions.

## Introduction

Gene regulatory networks (GRNs) modulated by intercellular signals control the generation of the specialized cell types that compose multicellular organisms [1,2]. These control

PRJEB5857, PRJEB6774, PRJEB27937, PRJEB32740 and PRJEB21105).

**Funding:** This research was funded by grants from the Medical Research Council UK (Mr/J003662/1, D.J.P and J.O.M.; Mr/N012291/1, D.J.P.), the Biotechnology and Biological Sciences Research Council UK (Bb/N006542/1, D.J.P. and J.O.M.), the Muir Maxwell Epilepsy Centre (http://www. muirmaxwellcentre.com/; M.I.D.) and Simons Initiative for the Developing Brain (https://sidb.org. uk/; grant number 529085, J.O.M. and D.J.P.), and Principal's Career Development and Edinburgh Global Research Scholarships from the University of Edinburgh (https://www.ed.ac.uk/institute-academic-development/postgraduate/doctoral/career-management/principals-scholarships; K.B. T.) and a Scholarship from the Ministry of Higher Education, Malaysia (https://www.mohe.gov.my/en; M.F.A.R.). The funders had no role in study design, data collection and analysis, decision to publish, or preparation of the manuscript.

**Competing interests:** The authors have declared that no competing interests exist.

**Abbreviations:** aCSF, artificial cerebrospinal fluid; AHP, afterhyperpolarization; AP, action potential; aP, atypical progenitor; aRGP, atypical RGP; BrdU, Bromodeoxyuridine; CP, cortical plate; CRC, Cajal–Retzius cell; CSF, cerebrospinal fluid; DAB, diaminobenzidene; DAPI, diamidino-2-phenylindole; dcKO, double conditional KO; DE, differentially expressed; DEA, differential expression analysis; DIG, digoxigenin; dLGE, dorsal LGE; DLN, deep layer neuron; DNP, dinitrophenol; EBSS, Earle's Balanced Salt Solution; EdU, ethynyldeoxyuridine; eGC, ectopic GABAergic cell; FACS, fluorescence-activated cell sorting; FBS, fetal bovine serum; FDR, false discovery rate; GABA, gamma aminobutyric acid; GE, ganglionic eminence; GFP, green fluorescent protein; GO, gene ontology; GRN, gene regulatory network; IP, intermediate progenitor; LFC, log2 fold change; MAST, Model-based Analysis of Single-cell Transcriptomics; MEM, minimum essential medium; Pax6 cKO, Pax6 conditional knockout; PCA, principal component analysis; PFA, paraformaldehyde; qRT-PCR, quantitative real-time polymerase chain reaction; RGP, radial glial progenitor; RMP, resting membrane potential; SAG, signaling agonist; scRNAseq, single-cell RNAseq; SLN, superficial layer neuron; SNN, shared nearest neighbor; TSA, Tyramide Signal Amplification; TSS, transcription start site; TTX, tetrodotoxin; UMAP, uniform manifold approximation and projection; UMI, unique molecular identifier.

mechanisms affect the developmental trajectories of cells in a variety of ways to guide the production of particular cell types and prevent the emergence of alternatives. Transcription factors whose levels vary among developing cells in precise, reproducible spatiotemporal patterns are essential components of GRNs. In some cases, their regional activation in response to inductive signals drives the production of region-specific cell types, but there are many other ways in which they can operate. For example, they can determine whether, and if so how, cells respond when confronted by inductive signals, i.e., their competence [3,4]. Restricting the competence of cells as they develop is likely to maximize the probability of them following reproducibly their stereotypical developmental trajectories, e.g., by mitigating the effects of biochemical noise in the signals they encounter or in the intracellular pathways processing those signals [5] and by preventing them responding in inappropriate ways to signaling molecules surrounding them.

The cerebral cortex is a complex amalgamation of 2 major neuronal cell classes generated by developmental cell lineages expressing different sets of transcription factors [6–9]. One cell class uses the excitatory neurotransmitter glutamate to propagate neuronal activity through cortical circuits and is produced by progenitors located in the developing cerebral cortex itself. It develops from cell lineages that express transcription factors including Pax6, Neurog2, and Eomes. The other cell type uses the inhibitory neurotransmitter gamma aminobutyric acid (GABA) to refine and elaborate patterns of cortical neuronal activity and is produced by progenitors located subcortically. It develops from cell lineages that express substantially different sets of transcription factors. Pax6 is one of the first transcription factors to be expressed differentially between the progenitors of excitatory and inhibitory cortical neurons [10–12], making it a good candidate to be involved in regulating the likelihood of cortical progenitors adopting an excitatory neuronal fate.

The *Pax6* gene emerged 500 to 700 million years ago and has been conserved through all triproblastic animal lineages, where it is involved in many neural and nonneural processes [13,14]. Its expression in the developing brain of extant vertebrates and invertebrates indicates that it acquired important functions very early in this organ's evolution. In mammalian embryos, it is activated prior to neural tube closure in the anterior neuroectoderm where brain forms [15]. Its importance for the production of cortical excitatory neurons is demonstrated by the phenotypes of constitutively mutant mouse embryos unable to make functional Pax6. These embryos show reduced cortical expression of genes involved in excitatory neuron production and increased cortical expression of genes involved in the development of subcortically derived cell types including inhibitory interneurons [16–24]. We set out to discover what *Pax6* does in cortical progenitors to help govern their normal production of excitatory neurons.

We began by examining the effects of inducing cortex-specific *Pax6* loss-of-function in cortical progenitors using population and single-cell transcriptomics followed by expression analysis of selected genes in tissue sections. The response was dichotomous: Many *Pax6*-null progenitors continued to generate excitatory neurons that made cortical layers relatively normally, while others adopted abnormal developmental trajectories, the nature of which varied with age and cortical location. Subsequent in vivo and in vitro experiments revealed that Pax6 blocks the deviant trajectories by reducing the ability of cortical cells to react abnormally to substances normally present—and carrying out other essential functions—around them. We propose that the main function of Pax6 in cortical development is to imbue the process with stability and reproducibility by protecting it from potentially destabilizing signals in the cortical environment.

## Results

### Removal of Pax6 from the progenitors of cortical neurons

Most cortical excitatory neurons are generated between embryonic day 12.5 (E12.5) and E16.5 in mice [25–29]. They are derived from cortical radial glial progenitors (RGPs), some directly and others indirectly via the initial production of transit-amplifying intermediate progenitors (IPs) [30–32]. All RGPs express Pax6 [11]. We used the *Emx1-Cre^ERT2* allele [33] to make tamoxifen-induced cortex-specific homozygous *Pax6* conditional knockouts (*Pax6* cKOs) (S1A Fig). Heterozygous littermates with deletion in just one *Pax6* allele served as controls; previous work on heterozygotes detected no abnormalities in cortical levels and patterns of Pax6 protein expression or cortical morphogenesis, almost certainly because known feedback mechanisms caused compensatory increases in Pax6 production from the normal allele [34–36]. When we gave tamoxifen at E9.5 (tamoxifen^E9.5), levels of normal *Pax6* mRNA in *Pax6* cKOs fell to <50% of control by E11.5, to approximately 10% of control by E12.5 and to almost zero by E13.5 (S1B Fig) and levels of Pax6 protein fell to approximately 5% of control by E12.5 (S1C and S1D Fig). By E12.5, Pax6 was undetectable by immunohistochemistry in almost all RGPs (except those in a narrow ventral pallial domain where *Emx1* is not expressed) (S1E and S1F Fig) while a Cre reporter, *RCE^EGFP* (S1A Fig; [37]), was active in most cortical cells (S1E Fig). Thus, tamoxifen^E9.5 ensured that the vast majority of cortical neurons was generated, directly or indirectly, from RGPs that had lost Pax6 protein.

### Pax6 loss caused ectopic gene expression in cortical cells

We first used bulk RNAseq to study the effects of tamoxifen^E9.5-induced *Pax6* cKO in rostral and caudal cortex at E12.5 and E13.5 (S2A Fig). Raw data are available at the European Nucleotide Archive accession numbers PRJEB5857 and PRJEB6774. We used 4 biological replicates for each location, age and genotype; principal component analysis (PCA) on all datasets taken together showed high-level clustering by age and location (S1G Fig).

The number of genes with significantly altered expression levels (adjusted $p < 0.05$) in *Pax6* cKO cortex increased approximately 3-fold between E12.5 and E13.5 (S2B Fig and S1 Table). At each age, the numbers of up-regulated and down-regulated genes were similar. We identified regulated genes with nearby Pax6 binding sites using published chromatin immunoprecipitation-sequencing data from E12.5 forebrain obtained by Sun and colleagues [38]. We followed their assignation of peaks to the gene with the nearest transcription start site (TSS), provided the peak lay within the genomic interval between 50 kb upstream of the TSS and 50 kb downstream of the transcription end site. The proportion of regulated genes with a nearby binding site was higher at E12.5 than E13.5 (S2C Fig), suggesting an accumulation of indirect gene expression changes with age.

We then examined which genes altered their expression levels in *Pax6* cKO cortex (S2D and S2E Fig). We found that a major effect was the ectopic activation of genes normally expressed only extracortically, either by surrounding noncortical telencephalic cells or by cells normally located outside the telencephalon (S2D and S2E Fig). Many of these genes encoded transcription factors known to be involved in cell specification [16,17,24,39,40–55]. Note that our study did not aim to provide new evidence on whether genes with altered cortical expression were normally directly regulated by Pax6 binding to their enhancers or promoters (for previous data on this, in addition to those used above in [38], see [19,56,57]).

In summary, these findings indicated that acute conditional cortex-specific Pax6 removal rapidly affected the specification of at least some embryonic cortical cells.

## Pax6 loss caused cortical cell lineage progressions to diversify

We used single-cell RNAseq (scRNAseq) to explore the lineage progression of cortical cells following Pax6 removal. We used the alleles described above (S1A Fig) and separated green fluorescent protein (GFP)-expressing cells by fluorescence-activated cell sorting (FACS) from single-cell suspensions of E13.5 and E14.5 *Pax6* cKO (tamoxifen$^{E9.5}$) and control rostral cortex before carrying out scRNAseq (Fig 1A). We focussed on rostral cortex, since it contained approximately 85% of changes detected by bulk RNAseq at E13.5 (S2B Fig). This gave 4 datasets of 6,266 cells from E13.5 *Pax6* cKO; 3,744 cells from E13.5 control; 4,259 cells from E14.5 *Pax6* cKO; and 4,137 cells from E14.5 control. Raw data are available at the European Nucleotide Archive accession numbers PRJEB27937 and PRJEB32740. Differential expression analysis (DEA) using scRNAseq data to calculate Pax6-loss-induced log$_2$ fold changes (LFCs) in average gene expression at E13.5 correlated well with LFCs detected by bulk RNAseq in rostral E13.5 cortex (S1H Fig), cross-validating data obtained from the 2 approaches.

At E13.5, comparison of *Pax6* cKO and control samples using uniform manifold approximation and projection (UMAP) dimensionality reduction indicated a high degree of similarity in their transcriptomic landscapes (Fig 1B). Graph-based clustering combined with analysis of the expression of cell type–selective marker genes (such as *Nes*, *Sox9*, *Hes5*, *Neurog2*, *Eomes*, *Fezf2*, *Sox5*, *Tbr1*, and *Calb2*; [51,52,58–65]) separated cells of both genotypes into recognized major classes: RGPs, IPs, deep layer neurons (DLNs), and Cajal–Retzius cells (CRCs) (Figs 1B, S3A, and S4). It also split the RGPs into 2 clusters, one of which contained very few cells in controls but many in *Pax6* cKOs (Figs 1B, S3B, and S4). We called these cells atypical RGPs (aRGPs) and explored their distinguishing features further.

The genes whose expression levels were shown by DEA to be the most different between RGPs and aRGPs are listed in S4 Table. Most gene ontology (GO) terms obtained by passing this list through the Database for Annotation, Visualization and Integrated Discovery v6.8 (DAVID v6.8; [66,67]) described processes involved in mitosis (S4 Table). For some genes, differences in their expression levels between RGPs and aRGPs might have been explained by the fact that a relatively higher proportion of aRGPs than RGPs were in S phase (S3C and S3D Fig). However, this was not the case for others, including some associated with GO terms describing cellular responses to extracellular factors, such as *Fos* (up-regulated) and *Hes5* (down-regulated) (S3A and S3F Fig and S4 Table). UMAP plots showed a tendency for aRGPs to have high *Fos* expression (S3F Fig), and immunohistochemistry revealed elevated Fos expression in E13.5 *Pax6* cKO cortex (S3G Fig). Changes in the expression of immediate early genes encoding AP-1 transcription factors of the Jun and Fos families, whose expression levels are known to be induced by a range of extracellular signals [68–73], suggested that the loss of Pax6 might have altered cellular responses to extracellular signals, an idea explored further below.

At E13.5, some RGPs, aRGPs, and IPs in *Pax6* cKOs showing ectopic activation of genes such as *Gsx2*, *Dlx1*, and *Dlx2* (selected as examples of genes normally expressed outside the cortex but within the telencephalon: "Tel" in S2E Fig) and *Prdm13* (an example of a gene normally expressed outside the telencephalon: "Extra-tel" in S2E Fig) (Fig 1C). This suggested that diversification of gene expression was occurring as cells progressed from the RGP to the IP identity. This was even clearer a day later.

At E14.5, UMAP dimensionality reduction followed by graph-based clustering combined with analysis of the expression of cell type–selective marker genes (including those used at E13.5 with the addition of layer markers such as *Cux2*, *Satb2*, and *Tle4*; [74–77]) separated cells of both genotypes into recognized major classes: RGPs, IPs, superficial layer 2/3 neurons (SLN-L2/3) and layer 4 neurons (SLN-L4), deep layer 5 neurons (DLN-L5) and layer 6

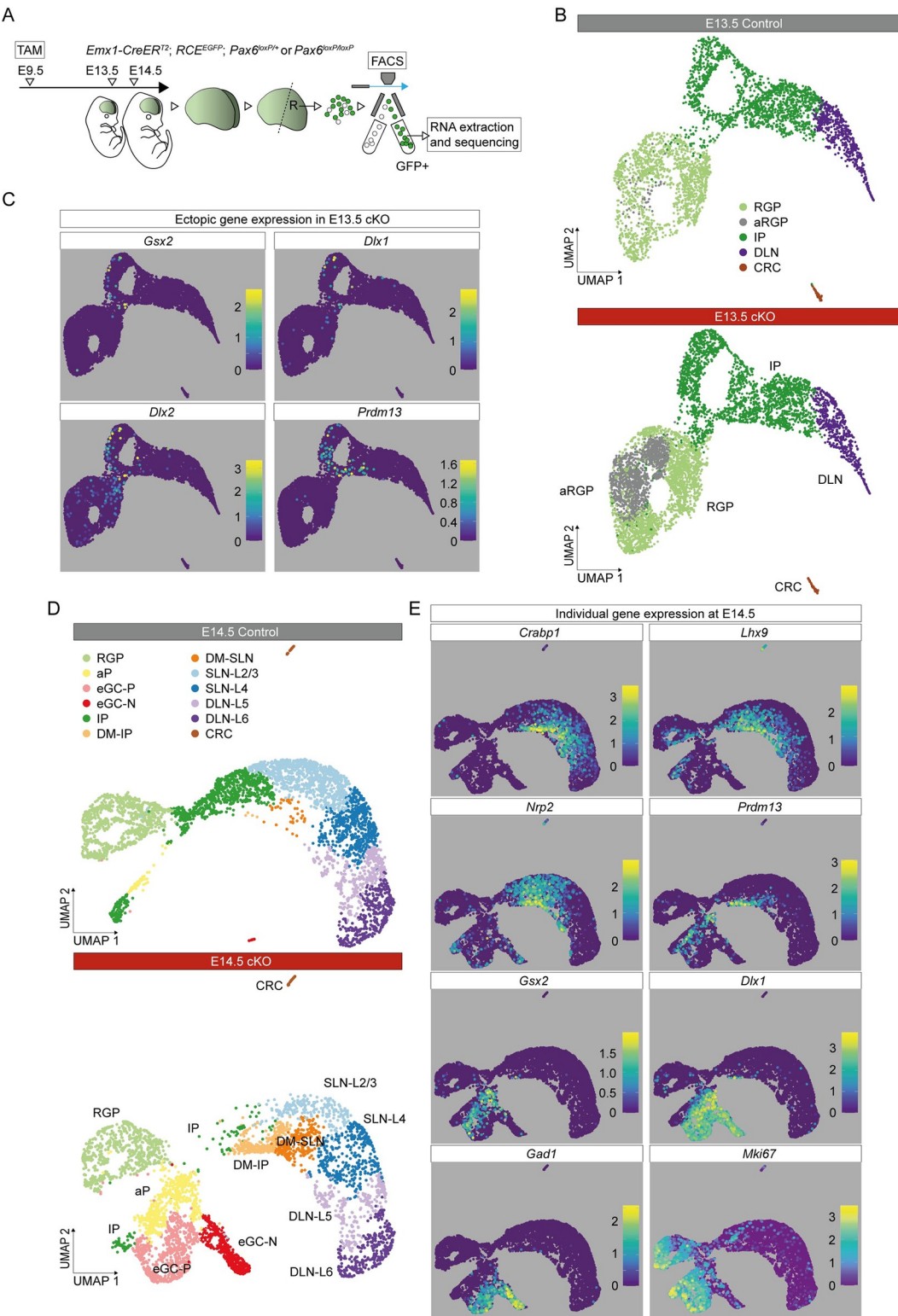

**Fig 1. Aberrant cell types and ectopic gene expression in *Pax6* cKO cortex.** **(A)** The experimental procedure: TAM was administered at E9.5; E13.5 and E14.5 rostral (R) cortex was dissociated into single-cell suspension; viable GFP+ cells were selected by FACS for scRNAseq. **(B)** UMAP plot of the scRNAseq data from *Pax6* cKO and control cells at E13.5. Data from the 2 genotypes were analyzed together and then split for visualization. **(C)** UMAP plots showing log₁₀ normalized expression of selected genes that were ectopically expressed in E13.5 *Pax6* cKO cortex. **(D)** UMAP plots of the scRNAseq data from *Pax6*

cKO and control cells at E14.5. Data from the 2 genotypes were analyzed together and then split for visualization. (**E**) UMAP plots showing $\log_{10}$ normalized expression of cell type–selective marker genes at E14.5. aP, atypical progenitor; aRGP, atypical RGP; CRC, Cajal–Retzius cell; DLN, deep layer neuron; DLN-L5 and DLN-L6, layer 5 or layer 6 deep layer neurons; DM-IP and DM-SLN, intermediate progenitor or superficial layer neuron in dorsomedial cortex; eGC-P and eCG-N, proliferating or nonproliferating ectopic GABAergic cells; FACS, fluorescence-activated cell sorting; IP, intermediate progenitor; *Pax6* cKO, *Pax6* conditional knockout; RGP, radial glial progenitor; scRNAseq, single-cell RNAseq; SLN-L2/3 and SLN-L4, layer 2/3 or layer 4 superficial layer neurons; TAM, tamoxifen; UMAP, uniform manifold approximation and projection.

neurons (DLN-L6), and CRCs (Figs 1D, S5A, and S6). The proportions of IPs and SLN-L2/3s were reduced in *Pax6* cKOs (S5B Fig), in line with previous reports [20,23,78,79].

Five additional clusters populated entirely, or very largely, by *Pax6* cKO cells had emerged. Two of them mapped between IPs and SLNs (Figs 1D and S6) and their cells expressed high levels of genes such as *Crabp1*, *Lhx9*, and *Nrp2* (Fig 1E), characteristic of cortical cells located relatively dorsomedially [80–82]. We designated cells in these 2 clusters dorsomedial (DM); the cells in one (DM-IPs) showed greater similarity to IPs than to SLNs (e.g., in terms of *Eomes* and *Neurog2* expression; S5A and S6 Figs), while cells in the other (DM-SLNs) were more similar to SLNs than to IPs (e.g., in terms of *Cux2* and *Satb2* expression; S5A and S6 Figs). Many DM cells expressed relatively high levels of *Prdm13* (Figs 1E and S6). Another 2 clusters contained cells expressing genes such as *Gsx2*, *Dlx*, and *Gad* family members, which are normally associated with the development of GABAergic interneurons (Figs 1E, S5A, and S6) [45,83–85]. We found that these cells mapped with GABAergic interneurons and the ganglionic eminence (GE) progenitors that generated them when we integrated our E14.5 *Pax6* cKO scRNAseq dataset with data from normal E13.5 and E14.5 ventral telencephalon [86] (S7 Fig). We called them ectopic GABAergic cells (eGCs).

A major difference between the 2 eGC clusters was that one (which we named eGC-P) showed strong expression of markers of proliferating cells (e.g., *Mki67*), while cells in the other (which we named eGC-N) did not (Figs 1E and S6). The fifth cluster contained proliferating cells (e.g., *Mki67*-expressing) that we called atypical progenitors (aPs). They were much more common in *Pax6* cKOs than in controls (S5B Fig). Their gene expression profiles suggested that they were intermediate between other types of cell in both controls and *Pax6* cKOs (Figs 1E, S5A, and S6). For example, they were RGP-like in expressing *Nes* and *Sox9* (albeit at lower levels in both genotypes) and IP-like in expressing *Eomes* (at lower levels in *Pax6* cKO cells). In *Pax6* cKOs, they were eGC-like in expressing *Gsx2*, *Dlx*, and *Gad* family members. Coexpression analysis revealed that small proportions of aPs in *Pax6* cKOs coexpressed a marker of cells undergoing normal cortical neurogenesis (*Neurog2* and *Eomes*) and a marker of eGCs (e.g., *Gsx2* and *Dlx1*) (S5C Fig).

Cells in all clusters expressed the telencephalic marker, *Foxg1* (S6 Fig). This indicated that cells undergoing ectopic activation of genes normally expressed in nontelencephalic tissue (e.g., *Prdm13*) did not lose entirely their telencephalic identity.

We next used RNA Velocity [87,88] to explore the direction and speed of movement of individual *Pax6* cKO cortical cells along their predicted developmental trajectories, with particular focus on aPs (Fig 2). For all control aPs and some *Pax6* cKO aPs, velocities were directed toward IPs (Fig 2A and 2B). Whereas some of these cells in *Pax6* cKO cortex expressed markers of normal cortical neurogenesis (e.g., *Neurog2* and *Eomes*), others expressed markers of GE-derived cells (e.g., *Gsx2* and *Dlx1*) (Fig 2C) and some were those shown previously to coexpress both (S5C Fig). This suggested that cells in the aP state were labile, with some transiently activating elements of the eGC expression profile before reverting to a more normal trajectory (further evidence for this is presented below). Other *Pax6* cKO aPs had velocities directed toward eGCs, either eGC-Ps or eGC-Ns (Fig 2B and 2C). These

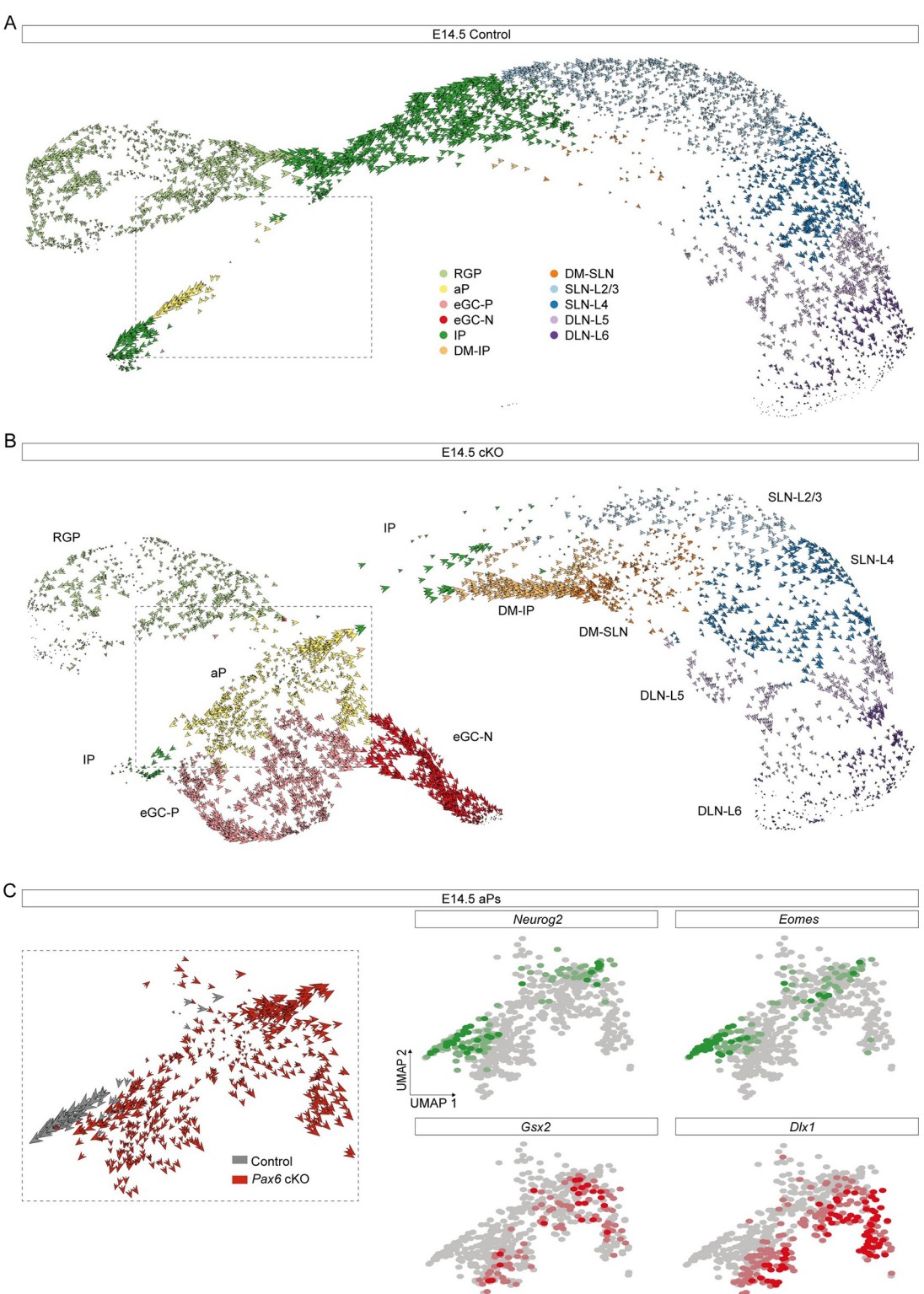

**Fig 2. Development of abnormal cell lineages in *Pax6* cKO cortex. (A, B)** RNA velocity analysis of individual E14.5 control and *Pax6* cKO cortical cells. Abbreviations: see Fig 1D. **(C)** Enlargement of the boxed area in (B) showing RNA velocity analysis of individual aP control and *Pax6* cKO cells and UMAP plots of *Neurog2*, *Eomes*, *Gsx2*, and *Dlx1* expression in aPs. aP, atypical progenitor; *Pax6* cKO, *Pax6* conditional knockout; UMAP, uniform manifold approximation and projection.

cells showed little or no expression of *Neurog2* and *Eomes* but strong expression of *Gsx2* and *Dlx1*, suggesting that they had become more highly committed to their aberrant fates.

In summary, Pax6 removal appeared to have 2 major effects. First, it expanded (i) the proportion of progenitor cells in a labile state between other more highly specified progenitor states (aPs) and (ii) the proportion of cells with a relatively dorsomedial cortical identity (DMs). Second, it diversified the set of cell lineage progressions open to cortex-born cells: While some lineages remained similar to those adopted by normal cortex-born cells, others resembled those normally followed by cells born outside the cortex.

## Spatiotemporal variation in the effects of Pax6 loss on selected gene expression

We next examined the effects of tamoxifen^E9.5-induced *Pax6* deletion on spatial and temporal patterns of expression of key Pax6-regulated genes in cortical sections using in situ hybridization and immunohistochemistry. Comprehensive visualizations of the expression patterns of selected genes were obtained by combining data from serial sections such as those in S8A Fig to generate surface-view reconstructions on representations of flattened cortical sheets (Fig 3; for reconstruction method, see S8B Fig).

*Ascl1* up-regulation began in lateral-most cortex and spread across its entirety between E13.5 and E14.5, while *Neurog2* was down-regulated in lateral cortex (except in the narrow ventral pallial domain where *Emx1^Cre* is not expressed) (Fig 3A–3C). Measurements of the proportions of cells expressing *Neurog2* or *Ascl1* with depth through the VZ and SVZ of lateral cortex showed that the distributions of *Neurog2*+ cells were replaced by similar distributions of *Ascl1*+ cells (S8C Fig). Medial cortex, on the other hand, maintained levels of *Neurog2* expression that were similar to control (Figs 3C and S8A) and contained a relatively high incidence of *Ascl1* and *Neurog2* coexpressing cells (S8A Fig). Eomes down-regulation was greater in lateral than in medial cortex (again, except in the narrow ventral pallial domain) (Fig 3D and 3E).

Tamoxifen^E9.5-induced *Pax6* deletion induced ectopic activation of *Prdm13* in a different pattern (Fig 3F and 3G). *Prdm13*+ cells were located in the medial two-thirds of the cortex, compatible with our scRNAseq analysis showing activation of *Prdm13* in DM cells (Fig 1E). They were mainly in the SVZ, where they intermingled with Eomes+ cells, some of which coexpressed both genes (Fig 3F). By E16.5, *Prdm13* expression remained detectable only in the most medial part of the cortex (S8D Fig).

These results indicated that Pax6 loss had distinct effects on the expression of different genes and that the effects varied with cortical region and age.

## Pax6 loss induced eGC production in a distinct spatiotemporal pattern

We next examined cells that deviated to the eGC fate by probing for expression of *Gsx2*, *Dlx1*, and *Gad1* (S9A–S9D Fig). In normal cortical development, *Gsx2* becomes active only in small numbers of late-stage (E16.5 or older) cortical SVZ cells that generate cell types other than cortical neurons [89] (these cells are seen in S9A Fig: "Control E16.5"). Following tamoxifen^E9.5, a wave of ectopic *Gsx2* activation was advancing rapidly across the cortex by E12.5. It began laterally and swept progressively further medially to occupy all parts of lateral cortex by E14.5 but did not extend all the way through medial cortex (Figs 3H and S9A). We examined the extent to which this change depended on when tamoxifen was administered (evidence in S10A and S10B Fig confirmed that tamoxifen administration at ages other than E9.5 also caused Pax6 removal from most RGPs within 3 d). We found similar distributions of Gsx2+ cells at E13.5 no matter whether tamoxifen was administered on E8.5, E9.5, or E10.5 (Figs 3H and S9A) and

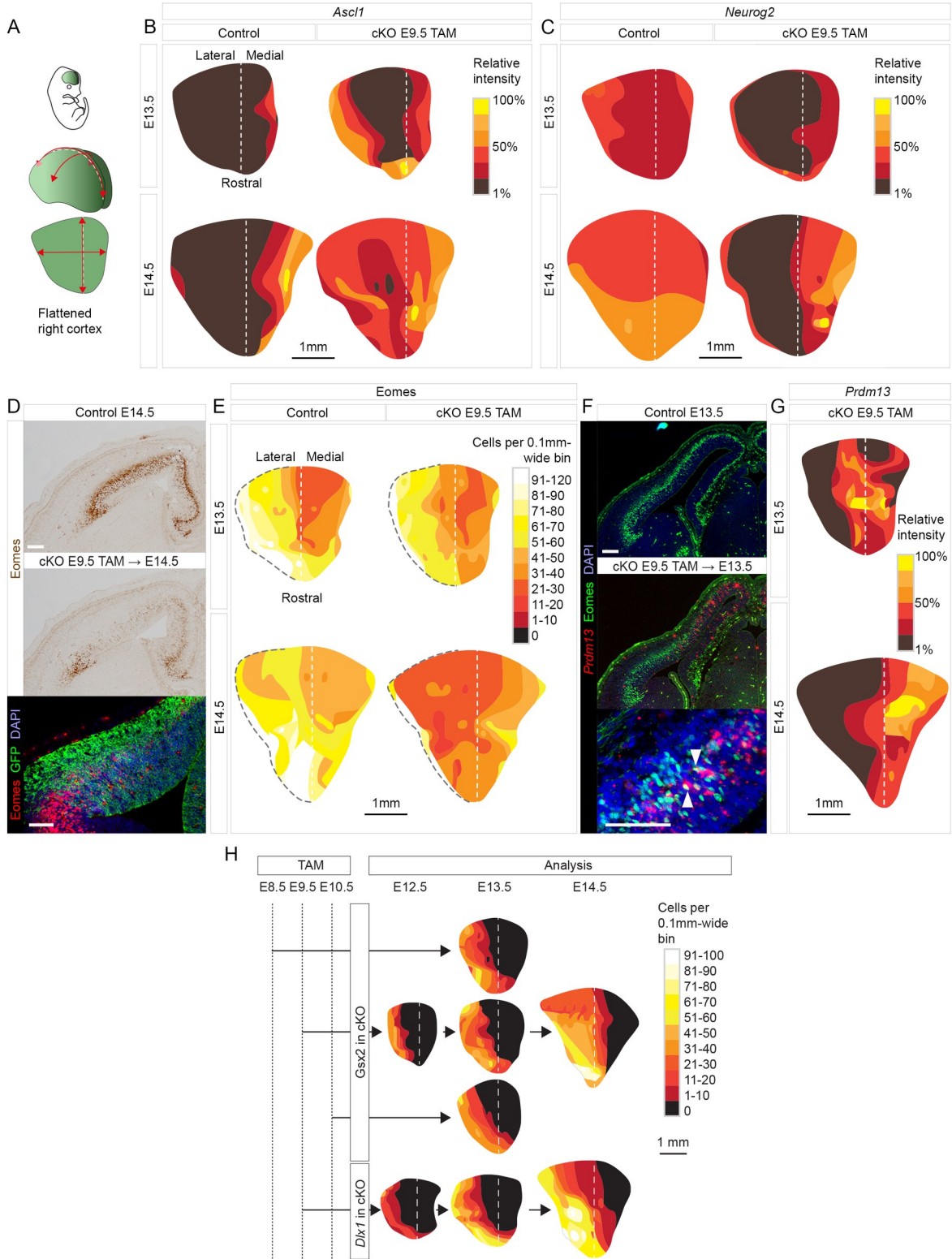

**Fig 3. Distinct spatiotemporal patterns of gene expression changes in *Pax6* cKO cortex. (A-C)** Flattened surface views of the cortex, oriented as in (A), to show the variation in the relative intensity of *Ascl1* and *Neurog2* staining across control and *Pax6* cKO cortex at E13.5 and E14.5. Method in S8B Fig. **(D)** Colorimetric and fluorescence immunoreactivity for Eomes in control and *Pax6* cKO cortex at E14.5 (with GFP labeling *Emx1*-lineage cells). Scale bars: 0.1 mm. **(E)** Flattened surface views of the cortex, oriented as in (A), to show the variation in the density of Eomes+ cells in control and *Pax6* cKO cortex at E13.5 and E14.5. Dashed black line: pallial–subpallial

boundary. **(F)** Immunoreactivity for Eomes and in situ hybridization for *Prdm13* in control and *Pax6* cKO cortex at E13.5. Arrowheads indicate double-labeled cells. Scale bars: 0.1 mm. **(G)** Flattened surface views of the cortex, oriented as in (A), to show the variation in the relative intensity of *Prdm13* staining in *Pax6* cKO cortex at E13.5 and E14.5. **(H)** Flattened surface views of the cortex oriented as in (A), showing the densities of Gsx2+ and *Dlx1*+ cells in control and *Pax6* cKO cortex at E12.5, E13.5 and E14.5 following tamoxifen administration at E8.5, E9.5, or E10.5. GFP, green fluorescent protein; *Pax6* cKO, *Pax6* conditional knockout.

even in E13.5 constitutive $Pax6^{-/-}$ mutants that had never expressed functional Pax6 (S9A Fig). When we administered tamoxifen later, on E13.5, Gsx2+ cells were distributed throughout the entire lateral cortex 3 d later. This resembled the distributions at similarly late ages (E14.5 to E16.5) following early tamoxifen administration (E8.5 to E10.5) and not the distributions 3 d after early tamoxifen administration (S9A Fig).

We concluded that the spatial distribution of Gsx2+ cells depended mainly on cortical age rather than time elapsed since Pax6 removal, suggesting that cortical factors that change with age have important influences on the outcome of Pax6 removal.

Tamoxifen$^{E9.5}$ induced a wave of ectopic *Dlx1* expression similar to that of Gsx2 expression, i.e., it was underway by E12.5 (S9B Fig) and had spread through lateral cortex but only encroached to a limited extent into medial cortex by E14.5 (Fig 3H). Tamoxifen$^{E9.5}$ also led to the generation of a large population of *Gad1*+ cells in the lateral cortex (S9C and S9D Fig). Most of these cells were cortically derived (i.e., they were GFP+ *Emx1*-lineage) but they were intermingled throughout their domain with other *Gad1*+ cells that were GFP-negative subcortically generated immigrants (arrows in S9D Fig).

In the VZ and SVZ of *Pax6*-deleted lateral cortex, *Gsx2*, *Dlx1*, and *Gad1* were activated by partially overlapping bands of cells centered progressively further basal to the ventricular surface (S9E Fig). The Gsx2+ and *Dlx1*+ bands overlapped the basal side of the Ascl1+ band and the *Dlx1*+ band extended further basally than the Gsx2+ band. This was followed by the *Gad1*+ band, which showed considerable overlap with the *Dlx1*+ band but less overlap with the Ascl1+ and Gsx2+ bands (summarized in S9F Fig). Where domains of expression overlapped, coexpressing cells were frequent. Small proportions of Gsx2+ or *Dlx1*+ cells coexpressed Eomes (arrows in S9G Fig), in agreement with findings in our scRNAseq data (Figs 2C and S5C).

We concluded that the production of eGCs unfolded in a distinct spatiotemporal pattern in mainly lateral cortex.

## Pax6 loss induced ectopic Olig2 expression largely independently of eGC production

We then examined the pattern of ectopic cortical activation of *Olig2*, which is expressed in progenitors that generate cortical interneurons and oligodendrocytes, is normally restricted to the embryonic subpallium at around E13.5 (S9H Fig: "Control E13.5") and later spreads as Olig2 + cells migrate into the cortex (S9H Fig: "Control E16.5") [33,90]. Our scRNA-seq data indicated that *Olig2* was not specifically marking eGCs but was expressed by many additional cell types including RGPs, aPs, IPs, and differentiating cells in *Pax6* cKOs (S6 and S10C Figs). Its ectopic spatiotemporal activation pattern differed from that of *Gsx2*, *Dlx1*, and *Gad1* to the extent that it appeared throughout the entire lateral cortex earlier, by E13.5, but was similar in showing relatively little activation in medial cortex, even at later ages (S9H Fig). The domain of Olig2 activation was similar in E13.5 to E16.5 embryos regardless of whether tamoxifen was given at E9.5, E10.5, or E13.5. In lateral cortex, many progenitors coexpressed *Olig2* and *Ascl1* (S9I Fig); this was supported by scRNAseq data showing that 51.8% and 67.3% of *Olig2*+ cells expressed *Ascl1* at E13.5 and E14.5, respectively. Nevertheless, our *Pax6* cKO E14.5 scRNAseq

data detected *Olig2* coexpression in only a small proportion (9.6%) of cells expressing eGC markers *Gsx2*, *Dlx1*, and *Gad1* (S10 Fig).

These findings suggested that the Pax6-loss-induced activation of *Olig2* and of eGC-expressed genes such as *Gsx2*, *Dlx1*, and *Gad1* occurred largely independently. They provided further evidence of spatiotemporal variation in the effects of Pax6 loss on the ectopic activation of different genes.

## The eGCs were highly proliferative

Our scRNAseq data indicated the existence of a substantial population of proliferating eGCs in E14.5 *Pax6* cKO cortex. This was demonstrated, for example, by the rising levels of the mitotic marker *Mki67* along the inferred pseudotime trajectory of the lineage leading to eGC-P generation (Fig 4A; trajectories were obtained using Slingshot and tradeSeq; [91–93]). To test this conclusion further, we used the *Emx1-Cre*$^{ERT2}$ allele with tamoxifen$^{E9.5}$ to delete *Pax6* and then labeled proliferating cells by administering the S phase marker 5-ethynyl-2′-deoxyuridine (EdU) at E13.5, 30 min before death (Fig 4B). We reacted sections for EdU and Gsx2, a marker of early eGCs (and also for GFP from a *Btg2*-GFP transgene that was incorporated into the mice for reasons given below) (Fig 4C). Most Gsx2+ cells were in S phase (mean = 59.0% ± 3.4 SD; counts were from 20 equally spaced coronal sections through the cortex for each embryo; $n$ = 5 embryos from separate litters; Sheet A in S3 Data), confirming their high level of proliferation.

We then studied the types of division that *Pax6* cKO cortical progenitors made. Previous work has shown that RGPs (Sox9+) and IPs (Eomes+) produce either postmitotic neurons or new progenitors [30,31,94,95]. Progenitors of the latter type, often described as proliferative progenitors, do not express the antiproliferative gene *Btg2*; others, often described as neurogenic, do express *Btg2* [96,97]. We used the *Btg2*-GFP transgene [97] with immunohistochemistry to identify neurogenic progenitors (Fig 4D and 4E). Many Gsx2+ cells expressed *Btg2* at E13.5 and E14.5, but a sizeable minority did not. Quantification in E13.5 tissue sections showed that 68.1% ± 6.5 (SD) of Gsx2 protein-expressing cells were also *Btg2*-expressing (counts were made in 20 equally spaced coronal sections through the cortex for each embryo; $n$ = 5 embryos from separate litters; Sheet A in S3 Data). This was similar to scRNAseq data, which showed *Btg2* expression in 76.2% and 74.0% of *Gsx2*+ cells at E13.5 and E14.5, respectively. These data indicated that, overall, about a quarter of the cortical cells that activated *Gsx2* were proliferative (i.e., *Btg2*-nonexpressing; their daughters would divide at least once more). The emergence in *Pax6* cKO cortex of substantial numbers of repeatedly and rapidly dividing progenitors caused a large expansion of the eGC population, described in the next section.

## Transient subcortical masses of eGCs formed beneath lateral cortex

Abnormal collections of *Gad1*+ cells coalesced beneath the cortical plate (CP) and superficial to the reduced population of Eomes+ cells in *Pax6* cKO lateral cortex between E14.5 and E16.5 (Figs 4F and S11A). We refer to them here as sub-CP masses. Their expression of GFP, which indicated they were *Emx1*-lineage (Figs 4G, S9C, and S9D), combined with their *Gad1* positivity identified these cells as eGCs (Fig 1D and 1E). Based on findings described above, it was likely that many of them had gone through an early transient phase of *Gsx2* expression (S9F Fig). To confirm this, we lineage-traced cells using *Gsx2-Cre* [33] and the GFP Cre-reporter allele [37].

Since we could not use *Gsx2-Cre* in combination with *Emx1-Cre*$^{ERT2}$, these experiments were done in *Pax6*$^{Sey/Sey}$ (*Pax6*$^{−/−}$) constitutive mutants (S11B–S11D Fig), whose pattern of Gsx2 expression was similar to that in *Pax6* cKO cortex (S9A Fig). We confirmed that *Gsx2*-lineage cells lost their Gsx2 protein as they moved away from the ventricular surface, activating the *Gsx2-Cre*-activated GFP reporter but no longer Gsx2 protein (S11 Fig). In E14.5 control

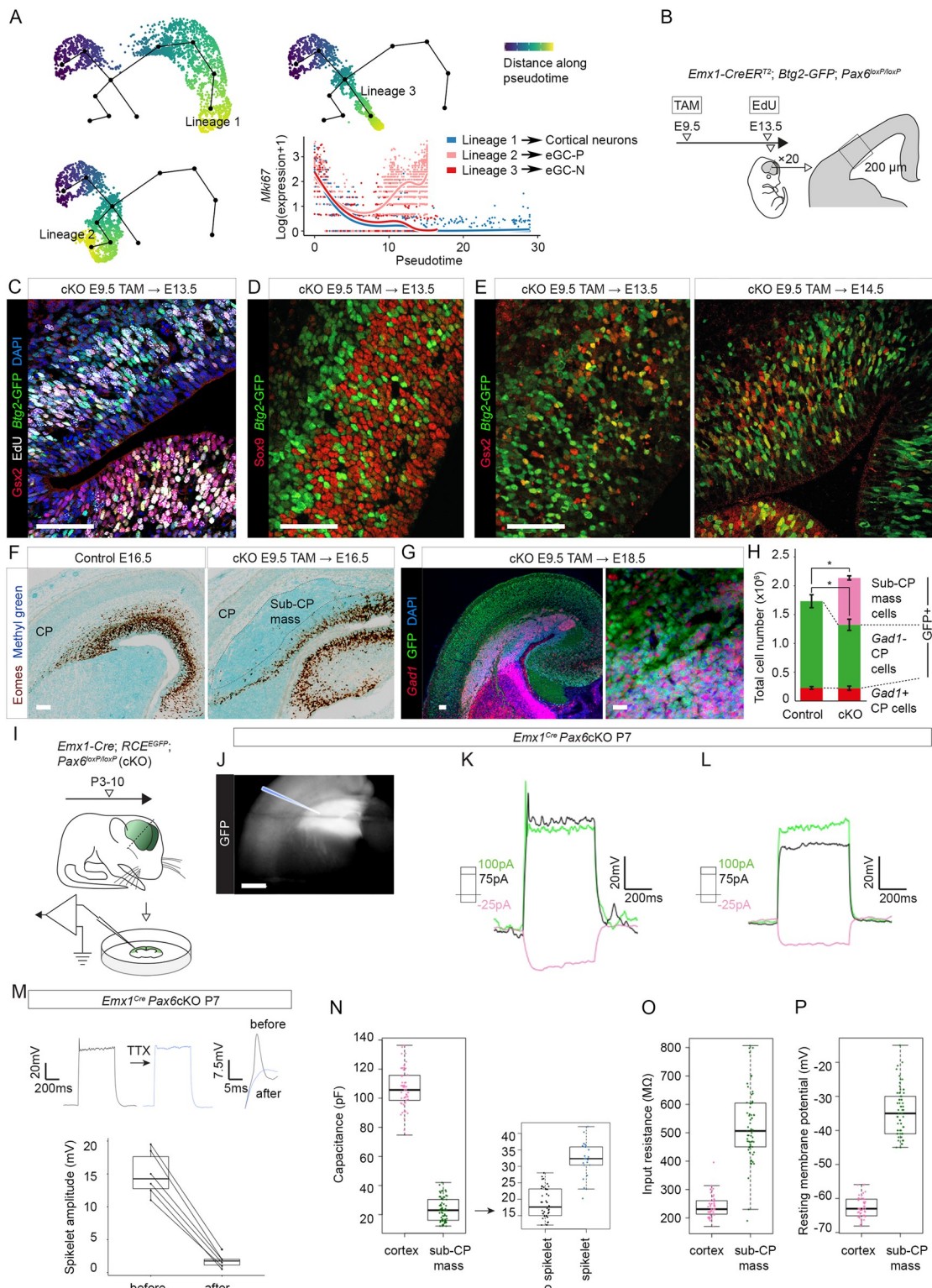

**Fig 4. The proliferation, coalescence, and electrophysiological properties of *Gsx2*-lineage eGCs. (A)** Major pseudotemporal trajectories inferred from E14.5 *Pax6* cKO scRNAseq data (one leading to eGC-Ps, one to eGC-Ns, and one to cortical glutamatergic neurons) and expression of the marker of proliferating cells, *Mki67*, along each. **(B)** The experimental procedure for EdU labeling. The *Emx1-Cre^ERT2* allele with tamoxifen^E9.5 was used to delete *Pax6* (embryos carried a *Btg2*-GFP transgene); EdU was given at E13.5, 30 min before death; 20 coronal sections equally spaced through the brain were immunoreacted for EdU, Gsx2,

and GFP; counts were made in the boxed area. **(C)** Fluorescence quadruple-staining for Gsx2, EdU, GFP (marking *Btg2*-expressing cells), and DAPI in E13.5 *Pax6* cKO cortex after the procedure in (B). Scale bar: 0.1 mm. **(D)** Fluorescence double-staining for Sox9 and GFP (marking *Btg2*-expressing cells) in E13.5 *Pax6* cKO cortex after tamoxifen[E9.5]. Scale bar: 0.1 mm. **(E)** Fluorescence double-staining for Gsx2 and GFP (marking *Btg2*-expressing cells) in E13.5 and E14.5 *Pax6* cKO cortex after tamoxifen[E9.5]. Scale bar: 0.1 mm. **(F)** Eomes immunoreactivity and methyl green counterstaining in control and E16.5 *Pax6* cKO cortex after tamoxifen[E9.5]. Scale bar: 0.1 mm. **(G)** Fluorescence immunoreactivity for GFP (*Emx1*-lineage) and in situ hybridization for *Gad1*+ cells in E18.5 *Pax6* cKO cortex after tamoxifen[E9.5]. Scale bars: 0.1 mm and 0.01 mm. **(H)** Quantifications of the total numbers of *Gad1*+ cells in the lateral CP (red: they were GFP-, subcortically derived), of *Gad1*− cells in the CP (green: GFP+, cortical-born) and of cells in the *Pax6* cKO sub-CP masses (pink: *Gad1*+, GFP+) in control and *Pax6* cKO E18.5 embryos after tamoxifen at E9.5 (for quantification method, see S11F Fig). Total numbers of cells were greater in *Pax6* cKO cortex ($p < 0.05$) and numbers of lateral CP cells were reduced ($p < 0.02$) (averages ± SEM; Student paired *t* tests; *n* = 4 embryos of each genotype, from 4 independent litters) (S4 Data). **(I)** The experimental procedure for electrophysiology (J-P). The *Emx1-Cre* allele was used to delete *Pax6*; embryos carried a GFP reporter transgene. Recordings were from sub-CP masses at P3–10. **(J)** Sub-CP mass in P7 slice prepared for electrophysiology: The cortex was GFP+ and the sub-CP mass was intensely so. Scale bar: 0.5 mm. **(K,L)** Examples of responses of sub-CP mass cells to current injections (square steps, magnitudes color-coded, 500 ms duration). Membrane voltages were held at −70mV. Some cells produced small spikelets (J), others did not (K). **(M)** TTX (300 nM) reduced spikelet amplitudes; examples of entire response and spikelet alone before and after TTX application; effects of TTX were significant ($p = 0.035$, Wilcoxon signed rank test; *n* = 6 cells) (Sheet A in S5 Data). **(N-P)** Passive electrical properties of P3-P10 sub-CP mass cells compared to P5-P7 cortical cells from layer 5 of somatosensory area 1 (*n* = 66 sub-CP mass cells, Sheet B in S5 Data; *n* = 49 cortex cells, data for these CP cells are in S5 Table). Sub-CP mass cells had significantly lower capacitance ($p = 2.2 \times 10^{-16}$, Mann–Whitney test) and significantly higher input resistance ($p = 2 \times 10^{-10}$, Mann–Whitney test) and resting membrane potential ($p = 2.2 \times 10^{-16}$, Mann–Whitney test). For capacitance, values were significantly higher among sub-CP mass cells that produced spikelets (*n* = 22 cells; *n* = 44 produced no spikelet) ($p = 1.9 \times 10^{-9}$, Mann–Whitney test). CP, cortical plate; EdU, 5-ethynyl-2′-deoxyuridine; GFP, green fluorescent protein; *Pax6* cKO, *Pax6* conditional knockout; TTX, tetrodotoxin.

cortex, *Gsx2*-lineage cells were scattered and had the elongated appearance associated with migration from the Gsx2+ subpallium (S11B Fig). In line with our prediction, most cells in the E16.5 sub-CP masses were *Gsx2*-lineage and most of their cells were *Gad1*+ (S11C Fig). In these experiments, we also observed a population of GFP+ *Gad1*-negative neurons scattered through the CP of E16.5 *Pax6*−/− but not control lateral cortex: Many of these neurons had the shape and apical dendrite associated with young cortical excitatory neurons (S11D Fig). This result agreed with a prediction from our scRNAseq analysis that some cells that first expressed markers of eGCs later reverted to a cortical excitatory neuronal fate (Fig 2C), reinforcing the suggestion of instability in the identities of *Pax6* cKO cells exiting the RGP state.

We returned to using *Pax6* cKOs (tamoxifen[E9.5]) to gain further information on the development of eGCs and sub-CP masses. Only extremely rarely did we find examples of *Emx1*-lineage (i.e., GFP+) *Gad1*+ cells in the CP of *Pax6* cKOs (an example is shown in S11E Fig), indicating that the vast majority of eGCs were unable to contribute to the CP. We estimated the total numbers of subcortically derived *Gad1*+ interneurons (i.e., non-*Emx1*-lineage, GFP-negative), the total numbers of cells contained in the sub-CP masses and the total numbers of all GFP+ cells in the lateral CP of control and *Pax6* cKO cortex at E18.5 (Fig 4F and 4H; methodology in S11F Fig). In *Pax6* cKOs, the numbers of *Gad1*+ GFP-negative cells in the lateral CP remained unchanged, indicating that immigration of subcortical *Gad1*+ cells into the lateral CP had proceeded normally. The numbers of GFP+ cells in the lateral CP were significantly reduced (Fig 4H; $p < 0.02$; Student paired *t* test). Adding the numbers of cells in the sub-CP masses to the numbers in the lateral CP revealed that, overall, significantly more cells in total were generated in *Pax6*-deleted than in control lateral cortex (Fig 4H; $p < 0.05$; Student paired *t* test). This was explicable by the switch of a significant proportion of cortical progenitors to the generation of highly proliferative eGCs that populated the sub-CP masses.

The sub-CP masses were no longer visible by postnatal day 34 (P34) (S11G Fig). This was most likely due to the death of their cells. The proportion of cells expressing the apoptosis marker caspase-3 was much higher in the sub-CP masses than in overlying CP (S11H Fig). It rose from 1.0% (±0.85 SD) at E14.5 to E16.5 to 7.1% (±4.6 SD) at P10, whereas it remained consistently very low in control cortex (mean = 0.29% ± 0.21 SD, all ages combined) (Sheet B

in S3 Data). We concluded that the very high level of proliferation among *Gsx2*-lineage cells in the lateral *Pax6* cKO cortex generated large sub-CP masses of eGCs that were eventually removed through cell death.

## Sub-CP mass cells showed immature electrophysiological properties

We tested whether sub-CP mass cells developed electrophysiological properties resembling those of interneurons by making whole-cell current-clamp recordings at P5 to P10 (Fig 4I) [98–100]. These ages encompassed those by which normal cortical neurons have acquired the ability to generate individual or trains of action potentials (APs) in response to depolarizing stimuli [101–103]. The sub-CP masses were easily identified in slices at all ages by their intense GFP expression (Fig 4J).

The properties of the sub-CP mass cells were similar across the range of ages studied here. None of them generated mature APs. A third (22/66) produced either spikelets (spikelet peak < 10 mV; spikelet amplitude = 5 to 25 mV; little or no afterhyperpolarization (AHP); Fig 4K) or, in 2 cases, underdeveloped APs (peak amplitude > 30 mV and AHP > 15 mV). Most (44/66) produced neither (Fig 4L). Spikelet amplitudes were reduced by approximately 90% following the addition of 300 nM tetrodotoxin (TTX), which blocks the voltage-gated $Na^+$ channels responsible for the rising phase of the AP [104,105] (Fig 4M), suggesting that spikelets were immature APs. One possibility was that the cells that produced spikelets were eGC-Ns, whereas those that did not were eGC-Ps.

The sub-CP mass cells had much lower capacitances and higher input resistances ($R_{in}$s) and resting membrane potentials (RMPs) than P5 to P7 cortical neurons recorded in layer 5 of primary somatosensory cortex (Figs 4N–4P and S14D–S14F and S5 Table). Their relatively low capacitances were a sign that they had relatively small somas (Fig 4N). When we split them into those that produced spikelets and those that did not, we found that the former had higher capacitances, indicating that they were slightly larger (Fig 4N). The relatively high $R_{in}$s and RMPs of the sub-CP mass cells, neither of which differed significantly between cells that did or did not generate spikelets, were likely attributable to immaturity in the numbers of ion channels in their cell membranes [103,106–108].

We concluded that although the transcriptomes of these cells showed progress toward a GABAergic interneuron fate, they were unable to develop corresponding cellular properties. Whether this was because they had a cell autonomous inability to mature and/or a problem with the environment in which they found themselves was not tested here.

## The production of eGCs did not depend on *Gsx2* activation

We next questioned whether early activation of *Gsx2* by eGCs contributed to their activation of genes such as *Dlx1* and *Gad1* and their repression of *Neurog2* and *Eomes*, as it does in the GEs [16,17,39,47]. We carried out tamoxifen^E9.5-induced *Pax6* and *Gsx2* cortex-specific codeletion (double conditional KO, or dcKO; Figs 5A and S12A). Codeletion of *Gsx2* did not prevent the production of *Dlx1*+ and *Gad1*+ cells in similar numbers as in *Pax6* single cKOs (Fig 5B and 5C). It had no detectable effect on activation of Ascl1 expression and did not reverse the Pax6-loss-induced loss of *Neurog2* from the bulk of the lateral cortex (S12B and S12C Fig). While Gsx2 protein was not detected in dcKOs (Fig 5B), mRNA from *Gsx2* exon 1 was (the deletion removed the homeodomain-encoding exon 2; S12A Fig; [47]), allowing us to use in situ hybridization to recognize cortical cells that had activated the *Gsx2* gene even in dcKOs. Using this approach, we found no evidence that loss of Gsx2 protein from *Gsx2*+ cells caused them to up-regulate Eomes expression (Fig 5D). Nor did it cause cortical activation of *Gsx1*, which occurs in the dorsal LGE (dLGE) following Gsx2 removal (S12D Fig) [17,52,109].

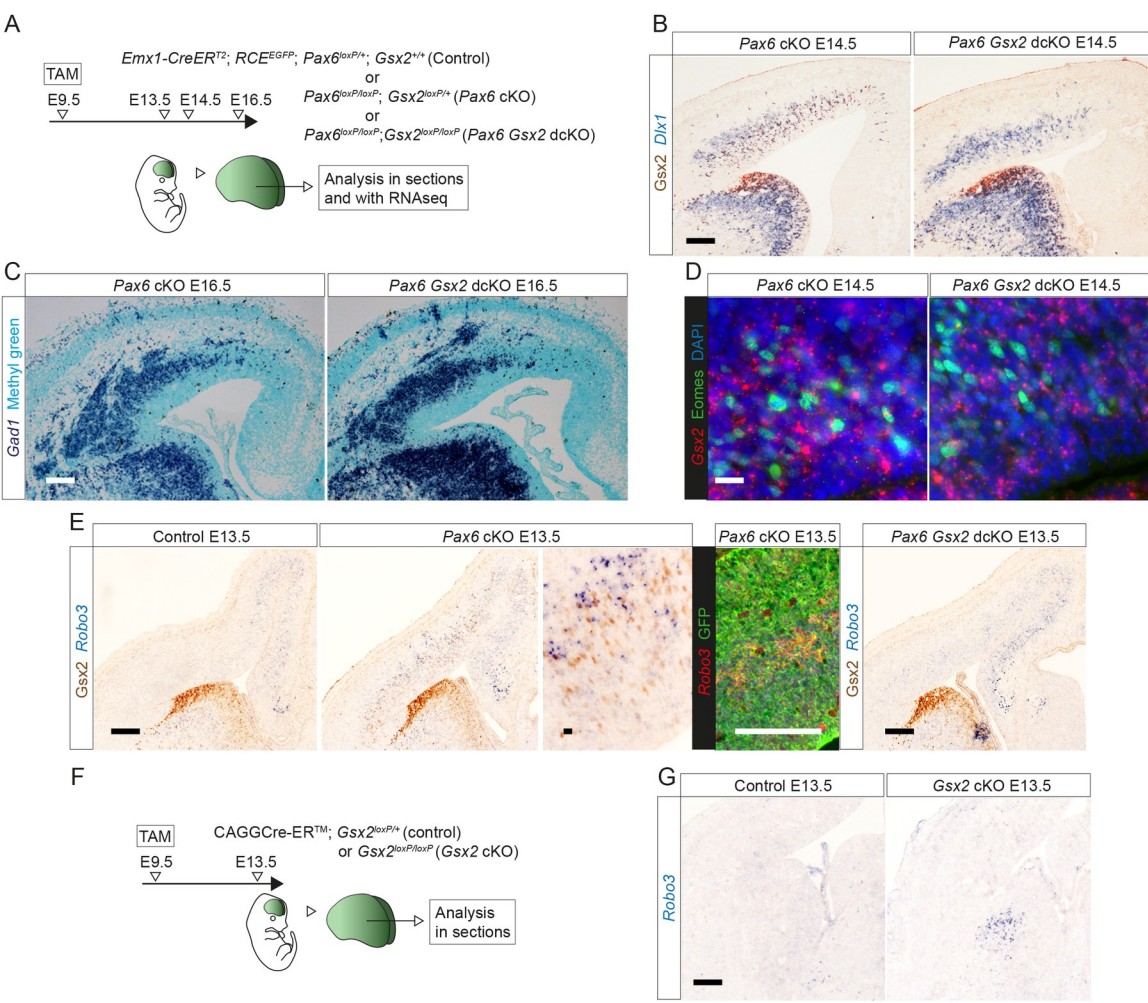

**Fig 5. The production of eGCs did not require Gsx2 activation.** (A) The experimental procedure for (B-E): TAM was administered at E9.5 to generate control embryos with functional alleles of both *Pax6* and *Gsx2*, a single cKO of *Pax6* or a dcKO of *Pax6* and *Gsx2*; brains were analysed at E13.5, E14.5, or E16.5. (B) Colorimetric immunoreactivity for Gsx2 and in situ hybridization for *Dlx1* in *Pax6* cKO and *Pax6 Gsx2* dcKO at E14.5. Scale bar: 0.1 mm. (C) Colorimetric in situ hybridization for *Gad1* in *Pax6* cKO and *Pax6 Gsx2* dcKO at E16.5. Scale bar: 0.1 mm. (D) Fluorescence in situ hybridization for *Gsx2* and immunoreactivity for Eomes in *Pax6* cKO and *Pax6 Gsx2* dcKO at E14.5. Scale bar: 0.01 mm. (E) Colorimetric and fluorescence in situ hybridizations for *Robo3*, colorimetric immunoreactivity for Gsx2, and fluorescence immunoreactivity for GFP in control, *Pax6* cKO, and *Pax6 Gsx2* dcKO at E13.5. Scale bars: 0.1 mm and 0.01 mm. (F) The experimental procedure for (G): TAM was administered at E9.5 to generate control embryos with a functional allele of *Gsx2* or a cKO of *Gsx2* throughout the embryo; brains were analysed at E13.5. (G) Colorimetric in situ hybridizations for *Robo3* in control and *Gsx2* cKO at E13.5. Scale bar: 0.1 mm. dcKO, double conditional KO; eGC, ectopic GABAergic cell; GFP, green fluorescent protein; *Pax6* cKO, *Pax6* conditional knockout; TAM, tamoxifen.

We obtained further evidence that Gsx2 loss had a very limited effect on the development of eGCs using RNAseq to compare gene expression levels in dcKOs versus *Pax6* single cKO cortex at E13.5. Raw data are available at the European Nucleotide Archive accession number PRJEB21105. This analysis found that only 2 genes were significantly up-regulated following deletion of Pax6 alone and significantly down-regulated by codeletion of Gsx2 (adjusted $p < 0.05$; S12E and S12F Fig), namely *Gsx2* itself and *Robo3*. Fifteen other genes showed small, significant differences in expression levels in dcKO compared to *Pax6* single cKO cortex: All were increased in dcKOs and only one of them, *Zic4*, was significantly affected (slightly increased) in *Pax6* single cKOs (S12F Fig).

We examined the effect of *Pax6* and *Gsx2* codeletion on *Robo3* expression more closely. In controls expressing both Pax6 and Gsx2, there was a declining medial-to-lateral gradient of *Robo3* expression across the cortex, similar to that described previously for Robo3 protein (Fig 5E) [110]. In *Pax6*-deleted cortex, there was additional *Robo3* expression in cells partly intermingled with, and partly basal to, the Gsx2+ cells in lateral cortex (Fig 5E). Few cells appeared to be double-labeled, agreeing with detection by scRNAseq of Gsx2 in only 0.2% of *Robo3* + cells at E13.5 and 6.2% at E14.5. These *Robo3*+ cells were cortex-born (i.e., they were GFP+) (Fig 5E). This additional *Robo3* expression in lateral cortex was reduced in *Pax6-Gsx2* dcKO cortex (Fig 5E).

We then examined how *Robo3* expression was affected by Gsx2 in its normal domain of expression in the GEs by using a *CAGG-Cre^{ERTM}* allele with tamoxifen$^{E9.5}$ to delete *Gsx2* throughout the entire embryo (Fig 5F). This increased *Robo3* expression in the LGE, where Gsx2 is normally strongly expressed. This effect was opposite to that caused by *Gsx2* deletion in eGCs (Fig 5G).

We concluded that although Gsx2 was one of the earliest genes expressed in eGCs, the expression of other eGC marker genes did not require its expression, suggesting that Pax6 deletion has parallel effects on multiple eGC marker genes. Moreover, the transcriptional responses of eGCs to Gsx2 loss were unlike those of normal Gsx2-expressing GE cells, which might reflect an intrinsic difference in the nature of the 2 cell types and/or a difference in their extracellular environments.

## Pax6 was not required in RGPs for their production of cortical excitatory neurons

As described above (S1B–S1F Fig), almost all Pax6 protein was lost across all *Emx1*-expressing cortex by E12.5 in *Pax6* cKOs generated by tamoxifen$^{E9.5}$. This near-universal loss had a near-universal effect on the expression of some genes. For example, some genes with strong expression across control E13.5 cortex became undetectable in most *Pax6* cKO cortical regions by E13.5 (S13 Fig), indicating that most cKO cells altered their gene expression at least to some extent. Nevertheless, Pax6 loss from RGPs did not stop them from generating large numbers of cells that were competent to migrate into the CP (in agreement with previous studies [11,78]) (S14A and S14B Fig), where they differentiated into deep and superficial layer neurons (our scRNAseq data; Fig 1D). Moreover, *Pax6* cKO CP contained Slc17a7 (Vglut1), a specific marker of glutamatergic neurons and synapses, distributed in a similar pattern to that in controls (S14C Fig).

Further evidence that Pax6 removal from RGPs did not prevent their generation of apparently normal cortical neurons came from whole-cell current-clamp recordings from GFP + cells in layers 2/3 and 5 in primary somatosensory cortex (S1) (S14D–S14H Fig). We detected no effects of genotype on the individual intrinsic functional properties of recorded cells (S5 Table), nor were cells separated by genotype using unsupervised hierarchical agglomerative clustering based on the cells' property profiles (S14H and S14G Fig; [111,112]). Of the 73 GFP+ cells recorded in *Pax6*cKO cortex, one (in layer 2/3) showed properties compatible with those of fast spiking interneurons (S14I Fig; [102,113]). It is possible that this represented a rare example of an eGC contributing to the cortical layers (see above; S11E Fig). No such cells were found in controls (*n* = 70 cells).

## Why *Pax6* deletion altered the fates of only some cortical cells: A hypothesis

We then turned to the question of why some cortical cells switched fate while others did not after Pax6 deletion from cortical progenitors. A parsimonious explanation was that Pax6 loss increased the potential for all RGPs to generate inappropriate cell lineages, but triggering this

required additional, extracellular factors. Systematic cross-cortical variations in the types and levels of these factors might have been responsible for generating the spatiotemporal patterns of normal and abnormal specification seen after *Pax6* deletion. We set out to test this idea.

## Immigrating cortical interneurons enhanced the misspecification of *Pax6* cKO cortical cells

The striking similarity between the spatiotemporal characteristics of the wave of eGC production and the wave of subcortically generated interneuron immigration (S15A Fig), which was not disrupted by Pax6 removal (see above), suggested that the immigrating interneurons might have been one source of extracellular factors triggering abnormal specification among *Pax6* cKO cortical cells. To test this possibility, we removed subcortical tissue from one side of cultured coronal slices of E13.5 *Pax6* cKO (tamoxifen$^{E9.5}$) telencephalon to prevent further interneuron influx and compared the production of Gsx2+ cells on the 2 sides after 48 h in culture, using the GFP reporter to mark cells of cortical origin (Fig 6A and 6B). The numbers of subcortically generated interneurons (i.e., GFP-negative *Gad1*+ cells) were approximately 4 times higher on the intact side (Fig 6C and 6D), as anticipated from previous work using this approach [6]. Proportions of GFP+ Gsx2+ cells were several times higher on the intact side, with significant differences in the more lateral parts of cortex (Fig 6E and 6F).

This outcome suggested that the proportions of *Pax6* cKO cortical cells that deviated to develop as eGCs was influenced by extracellular factors.

## Misspecification of *Pax6* cKO cortical cells depended on their ability to respond to Shh

We then hypothesized that the signaling molecule, Shh, might be one factor contributing to the reprogramming of *Pax6*-deleted RGPs and their daughters. The embryonic cortex contains Shh from a variety of sources, including immigrating interneurons [114–116] and cerebrospinal fluid (CSF) [117]. Immunohistochemistry showed that Shh levels varied considerably with cortical location and that its distribution patterns were similar in control and *Pax6* cKO embryonic cortex of equivalent ages (Figs 6G and S15C; evidence for antibody specificity is in S15B Fig). Shh levels were higher laterally at E13.5 and increased across the cortex over the following 2 d (Fig 6G).

To test the importance of endogenous Shh, we injected either an antagonist of the Shh receptor Smo (vismodegib; [118]) into the lateral ventricle or electroporated a plasmid expressing both an shRNA against *Smo* and GFP into the cortex of *Emx1-Cre*-induced *Pax6* cKO embryos and measured the effects on cortical Gsx2 expression (Fig 6H–6M). (We used *Emx1-Cre* rather than *Emx1-Cre*$^{ERT2}$ because we found that it gave better survival rates following in utero surgery, while inducing a similar pattern of Gsx2 expression.) Vismodegib intraventricular injection significantly lowered by approximately 40% the proportions of proliferative zone cells expressing Gsx2 compared to vehicle-only injection (Fig 6H–6J). Cells expressing *Smo* shRNA (GFP+) were on average significantly less immunoreactive for Gsx2 than a randomly selected sample of interspersed nonexpressing (GFP−) cells, a difference that was lost when a control scrambled shRNA (GFP+) was used (Fig 6K–6M; for quantification method, see S15D Fig).

Further evidence came from adding the Shh pathway blocker cyclopamine, either in beads or in solution, to cultured *Pax6* cKO slices (Fig 6N and 6O). This reduced their ectopic cortical expression of Gsx2. Interestingly, it had little if any effect on normal Gsx2 expression in the GEs, suggesting that their state of commitment was higher than that of the more labile eGC population.

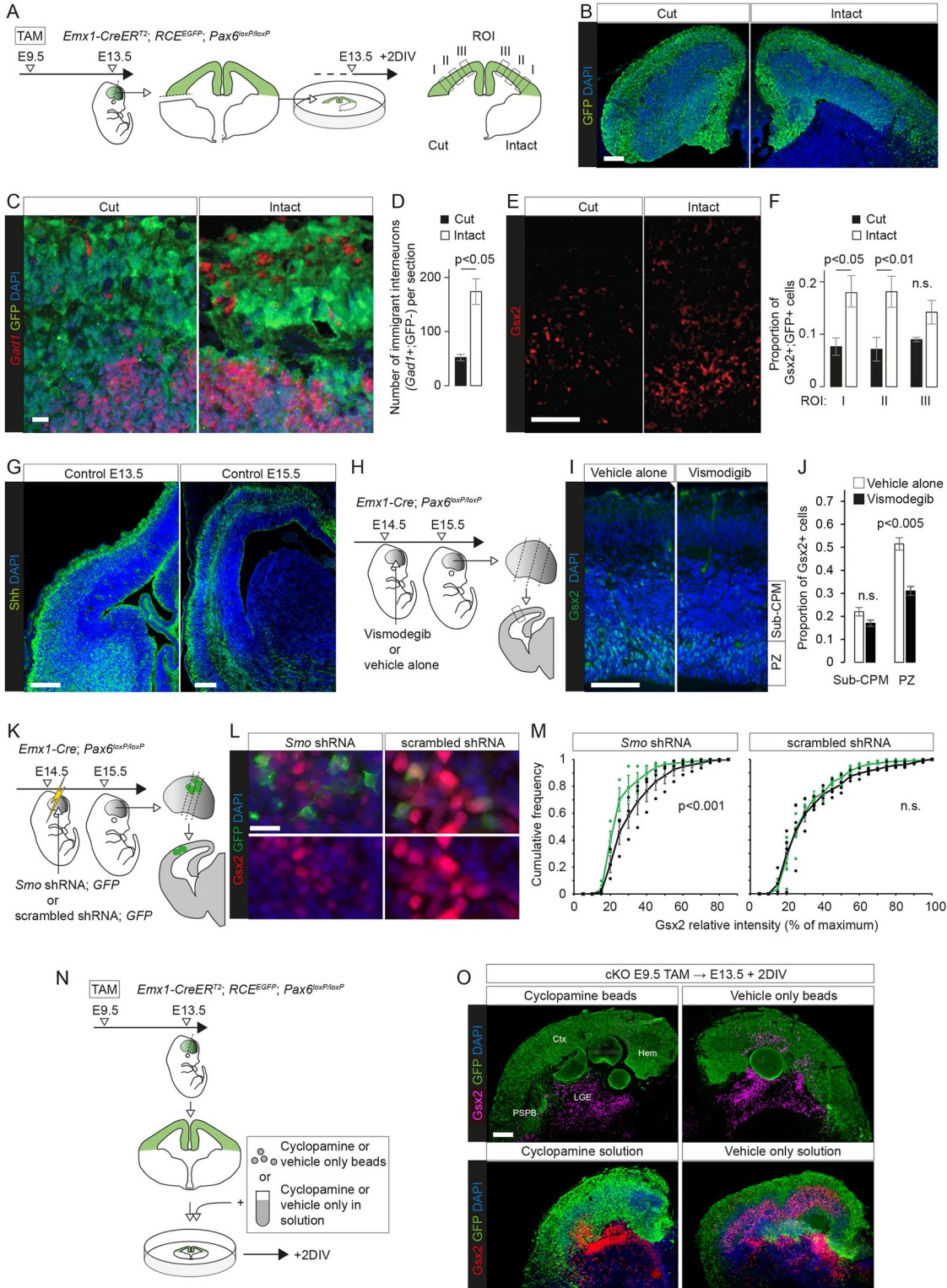

**Fig 6. Extracellular signals promoted eGC production in *Pax6* cKOs cortex. (A)** The experimental procedure for (B-F): TAM was given at E9.5 to generate *Pax6* cKOs, with Cre-deleted cells expressing GFP; coronal slices were cultured on E13.5 with the ventral telencephalon removed on one side; after 2 DIV, sections from cultured slices were cut and processed. Gsx2+ GFP+ cells were counted in 3 ROIs on each side. **(B, C)** GFP immunoreactivity and in situ hybridizations for *Gad1* in sections prepared as in (A). Scale bars: 0.1 mm and 0.01 mm. **(D)** Average (±SEM) numbers of immigrant *Gad1*+ interneurons (i.e., GFP nonexpressing) per

section were lower on the side lacking ventral telencephalon ($n = 3$ independent cultures; Student paired $t$ test) (Sheet A in S6 Data). **(E)** Gsx2 immunoreactivity in sections prepared as in (A). Scale bar: 0.1 mm. **(F)** Average (±SEM) proportions of GFP+ cells that were Gsx2+ in each ROI in (A) ($n = 3$ independent cultures; Student paired $t$ tests; n.s., not significant) (Sheet B in S6 Data). **(G)** Immunoreactvity for Shh in control telencephalic sections at E13.5 and E15.5 (see S15B Fig for evidence of antibody specificity). Scale bar: 0.1 mm. **(H)** The experimental procedure for (I, J): Vismodegib or vehicle alone was injected into the ventricle of E14.5 *Pax6* cKO embryos made using *Emx1-Cre*; central regions of coronal sections at 3 rostral-to-caudal levels were analysed at E15.5. **(I)** Gsx2 immunoreactivity in boxed region in (H). Scale bar: 0.1 mm. **(J)** Average (±SEM) proportions of cells in the PZs and CPMs that were Gsx2+ ($n = 5$ embryos from 3 litters given vehicle alone; $n = 6$ embryos from 3 litters given vismodegib; Student $t$ tests) (Sheet A in S7 Data). **(K)** The experimental procedure for (L, M): Constructs expressing *Smo* shRNA + GFP or scrambled shRNA + GFP were electroporated into the cortex of E14.5 *Pax6* cKO embryos made using *Emx1-Cre*; electroporated cells were analysed at E15.5 (as in S15D Fig). **(L)** Gsx2 and GFP immunoreactivity in electroporated regions. Scale bar: 0.01 mm. **(M)** Cumulative frequency distributions of the intensity of Gsx2 immunoreactivity in electroporated cells (GFP+; green) and surrounding randomly selected non-electroporated cells (GFP−; black) for the 2 constructs (see Figs 6K and S15D) ($n = 3$ embryos from 3 litters given *Smo* shRNA; $n = 4$ embryos from 3 litters given scrambled shRNA; Kolmogorov–Smirnov tests) (Sheet B in S7 Data). **(N)** The experimental procedure for (O): TAM was given at E9.5 to generate *Pax6* cKOs and Cre-deleted cells expressed GFP; coronal slices of telencephalon were cultured on E13.5; cyclopamine or vehicle alone were added either on beads or in solution (10 μM); slices were cultured for 2 DIV. **(O)** Sections from cultured slices obtained as in (N) were immunoreacted for Gsx2 and GFP. Scale bar: 0.1 mm. Ctx, cortex; DIV, day in vitro; GFP, green fluorescent protein; Hem, cortical hem; LGE, lateral ganglionic eminence; *Pax6* cKO, *Pax6* conditional knockout; PSPB, pallial–subpallial boundary; PZ, proliferative zone; ROI, region of interest; sub-CPM, sub-cortical plate mass; TAM, tamoxifen.

Previous work had shown that the ability of embryonic telencephalic cells to express GE marker genes in response to the ventralizing morphogen Shh requires the transcription factor Foxg1, which we found was expressed by cortical RGPs and their daughters in both control and Pax6 cKO cortex (S6 Fig) [119,120]. We postulated that Pax6 and Foxg1 have opposing actions (which might be direct, indirect, or both) on some aspects of cortical cells' competence to respond to Shh, including their ability to activate ventral telencephalic marker genes, but not others (Fig 7A). This idea was based on the following evidence from our RNAseq data and previous studies. First, we found that Pax6 removal caused little or no change in canonical readouts of Shh activity, namely *Ptch1* and *Gli1* mRNA expression levels [121–126]; only *Gli1* was significantly up-regulated to a small extent (LFC = 0.56) in caudal cortex at E13.5 (S1 Table). Second, in *Pax6* cKO cortex, there were no abnormalities in the expression of mRNAs for Shh itself, the Shh receptor Smo, or modulators of the Shh intracellular signal transduction pathways such as Kif7 and Sufu [127–129] (S1 Table). Third, previous work in *Foxg1*$^{-/-}$ telencephalon found that while cells failed to activate GE marker genes in response to Shh, they did activate *Ptch1* and *Gli1* normally [120].

To test the prediction that the up-regulation of GE markers in *Pax6* cKOs would be reversed by Foxg1 removal, we used tamoxifen$^{E9.5}$-induced *Emx1-Cre*$^{ERT2}$ to delete both copies of *Pax6* together with both, one, or neither copies of *Foxg1* from embryonic cortical cells (Figs 7B and S16A). Deletion of both copies of *Foxg1* resulted in the loss of *Foxg1* mRNA from almost all cortical cells by E13.5 (S16B and S16C Fig); the few remaining undeleted cells formed small clones expressing both Foxg1 and Pax6 (arrows in S16C Fig). Deletion of one copy of *Foxg1* appeared to lower its cortical mRNA and protein levels (S16C Fig).

Deletion of both copies of *Pax6* together with one copy of *Foxg1* reduced the numbers of cortical cells expressing Gsx2, *Dlx1*, and *Gad1* at E14.5 (Fig 7C and 7D). Deletion of both copies of both *Pax6* and *Foxg1* abolished cortical expression of Gsx2 and *Dlx1* (Fig 7C and 7D) and left only *Gad1*+ cells that were GFP-negative and presumably had originated subcortically (arrows in Fig 7E). Deletion of one or both copies of *Foxg1* also reduced Ascl1 expression in *Pax6* cKOs, with a greater effect in lateral than in medial cortex (S16D Fig). The reduction of the proportions of progenitor layer cells that were Eomes+ in E16.5 *Pax6* cKO cortex was reversed by codeletion of one or both copies of *Foxg1* (S16E and S16F Fig).

To confirm that these actions of Foxg1 were cell autonomous, we electroporated *Pax6*$^{-/-}$; *Foxg1*$^{-/-}$ dcKO cortex with a plasmid construct that resulted in the expression of mCherry and

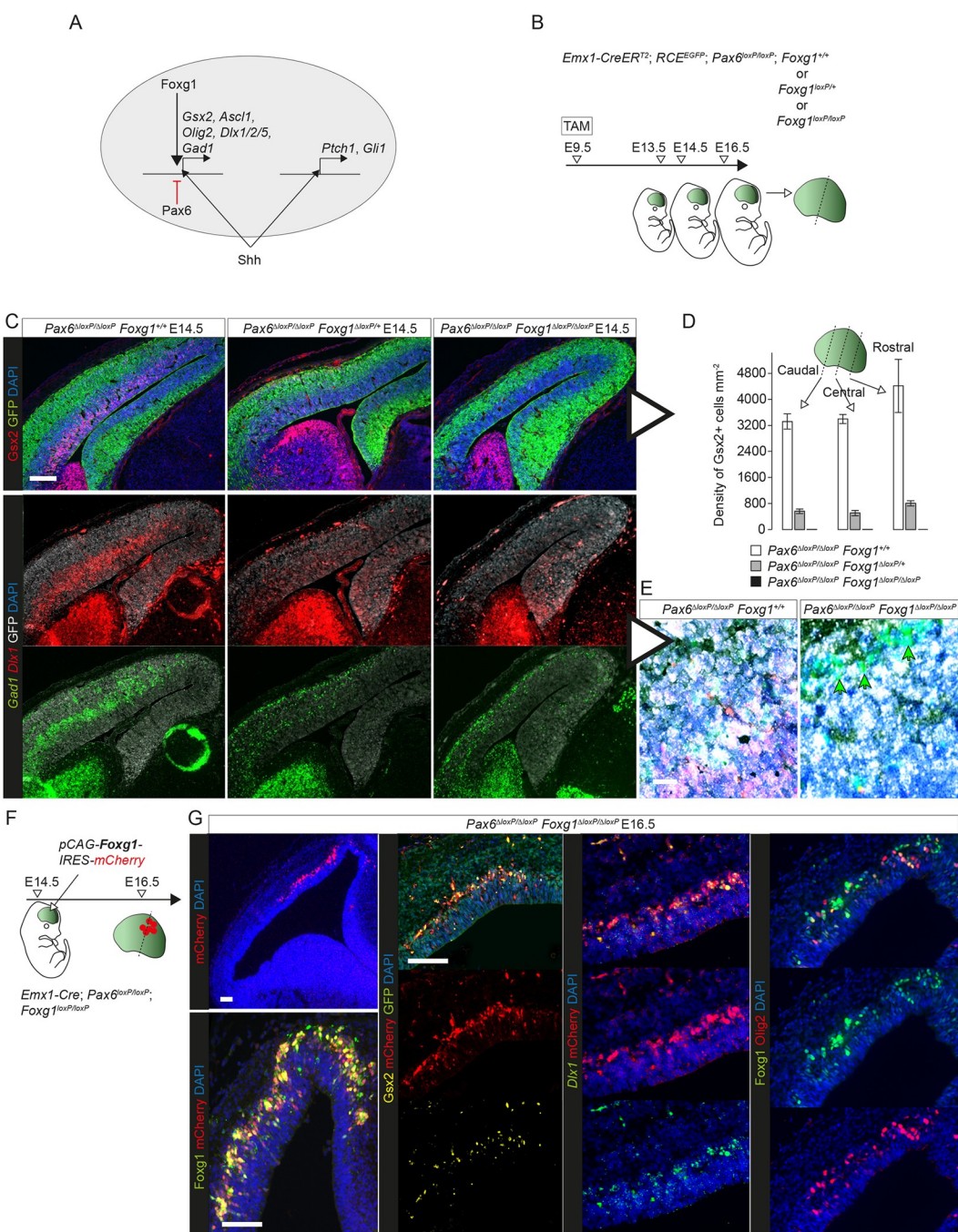

**Fig 7. Foxg1 required cell autonomously for eGC production. (A)** A hypothesis: Pax6 suppresses specifically Foxg1-permitted Shh-induced generation of eGCs without interfering with other effects of Shh pathway activation in these cells. **(B)** The experimental procedure for (C–E): TAM was administered at E9.5 to generate *Pax6* cKOs in which neither, one, or both *Foxg1* allele(s) were also deleted (for alleles, see S16A Fig); brains were analysed at E13.5, E14.5, or E16.5. **(C)** Immunoreactivity for Gsx2 and GFP and in situ hybridizations for *Gad1* and *Dlx1* in E14.5 cortex from *Pax6* cKO embryos in which neither, one, or both *Foxg1* allele(s) were deleted by TAM at E9.5. Scale bar: 0.1 mm. **(D)** Densities of Gsx2+ cells in the lateral cortex of E14.5 embryos with the 3 genotypes in (C) (averages ± SEM; *n* = 3 embryos of each genotype, from 3 independent litters) (Sheet C in S7 Data). **(E)** High magnification images from (C): At least the majority of residual *Gad1*+ (green) cells in *Pax6 Foxg1* double KOs (arrows) were GFP-negative (i.e., not white) of subcortical origin (non-*Emx1*-lineage). Scale bar: 0.01 mm. **(F)** The experimental procedure for (G): *Pax6 Foxg1* dcKO cortex made using *Emx1-Cre*, avoiding the need for TAM; a construct expressing Foxg1 and mCherry was electroporated into the cortex on E14.5; coronal sections were analysed on E16.5. **(G)** Results of experiment in (F): expression and coexpression of Foxg1, Gsx2, Olig2, mCherry, and GFP protein and *Dlx1* mRNA in coronal sections. Scale bars: 0.1 mm. dcKO, double conditional KO; eGC, ectopic GABAergic cell; GFP, green fluorescent protein; *Pax6* cKO, *Pax6* conditional knockout; TAM, tamoxifen.

the reexpression of Foxg1 by small groups of cells (Fig 7G and 7F). Two days after electroporation, an average of 33.2% (±7.4 SD; n = 3 embryos) of electroporated cells reexpressed Gsx2 (Sheet E in S7 Data) and many were reexpressing *Dlx1* and Olig2 (Fig 7G). Electroporated cells were much less likely than their non-electroporated neighbors to express Eomes (S16G Fig), in line with their redirection to an eGC-like fate.

In summary, these findings all suggested that Pax6 limits the competence of cortical cells to respond to Shh in their local environment by preventing them from adopting Foxg1-dependent developmental trajectories toward GE-like fates.

## *Pax6* deletion increases cortical cells' sensitivity to Shh pathway activation

We then compared the sensitivity of control and *Pax6* cKO cortical cells, in terms of their ability to express GE/eGC marker genes in response to Shh pathway activation. We dissociated E13.5 control or *Pax6* cKO cortex (tamoxifen$^{E9.5}$) carrying the GFP Cre-reporter, cultured the cells for 48 h in the presence of Shh signaling agonist (SAG; [130]) or vehicle alone (Fig 8A) and quantified numbers of GFP+ cells expressing Gsx2, Olig2, or *Dlx1* (Fig 8B; quantification method in S17 Fig). In common with previous studies, we used SAG concentrations in the nM range [130,131], which existing evidence suggested would likely have covered the levels of pathway activation experienced by telencephalic cells in vivo [117,132].

We found that no cells cultured with the lowest doses of SAG expressed Gsx2, *Dlx1*, or Olig2, despite the fact that all 3 genes would have been expressed by significant numbers of cells in E13.5 *Pax6* cKO cortex. The likely explanation for this, in line with suggestions made above, was that they were in a labile state requiring continual activation of their Shh signaling pathways to maintain their aberrant identity, and the signals they were receiving in vivo would been have dissipated by dissociation.

We found that control cells responded in a concentration-dependent manner to addition of SAG, in agreement with previous work showing that nonphysiological elevation of Shh signaling in normal embryonic cortex can activate the expression of ventral telencephalic marker genes [39,133–137]. However, *Pax6* cKO cells were significantly more sensitive to SAG than control cells (Fig 8C). The concentration-response functions for Gsx2 and *Dlx1* were relatively similar, reflecting the close association between their expression patterns in vivo, but differed from those for Olig2, which showed a different pattern of activation in vivo (S9H Fig). EC$_{50}$s for Olig2 were approximately 2 to 3 times higher, with cells less likely to express Olig2 than Gsx2 or *Dlx1* in response to low/intermediate levels of SAG (Fig 8C). Since our in vivo findings had shown that early Olig2 activation was more widespread than early Gsx2 and *Dlx1* activation in *Pax6* cKO cortex (S9A, S9B, and S9H Fig), this suggested that factors additional to Shh activation were required to explain the difference between the in vivo patterns of activation (see the next section).

The concentration-response functions for Gsx2 and *Dlx1* appeared to plateau with approximately 85% of GFP+ cells expressing the markers, suggesting that approximately 15% of E13.5 cortical cells were not competent to respond to Shh activation. Since this was similar to the percentage of differentiating glutamatergic neurons in E13.5 control or *Pax6* cKO cortex (S3B Fig), we tested whether the incompetent cells were those that were most highly differentiated. In one set of experiments, we identified differentiating neurons by their expression of Tubb3 (Fig 8D and 8E). We found that 9.9% to 11.5% of control and *Pax6* cKO GFP+ cells expressed Tubb3 whether SAG was added or not and that all GFP+ Gsx2-negative cells in SAG-treated cultures were Tubb3+, with only 2.0% to 2.3% of GFP+ cells expressing both Gsx2 and Tubb3. In another set of experiments, we identified cells that had divided in culture by adding the thymidine analogue EdU to the culture medium (Fig 8A, 8F, and 8G). We found that EdU was

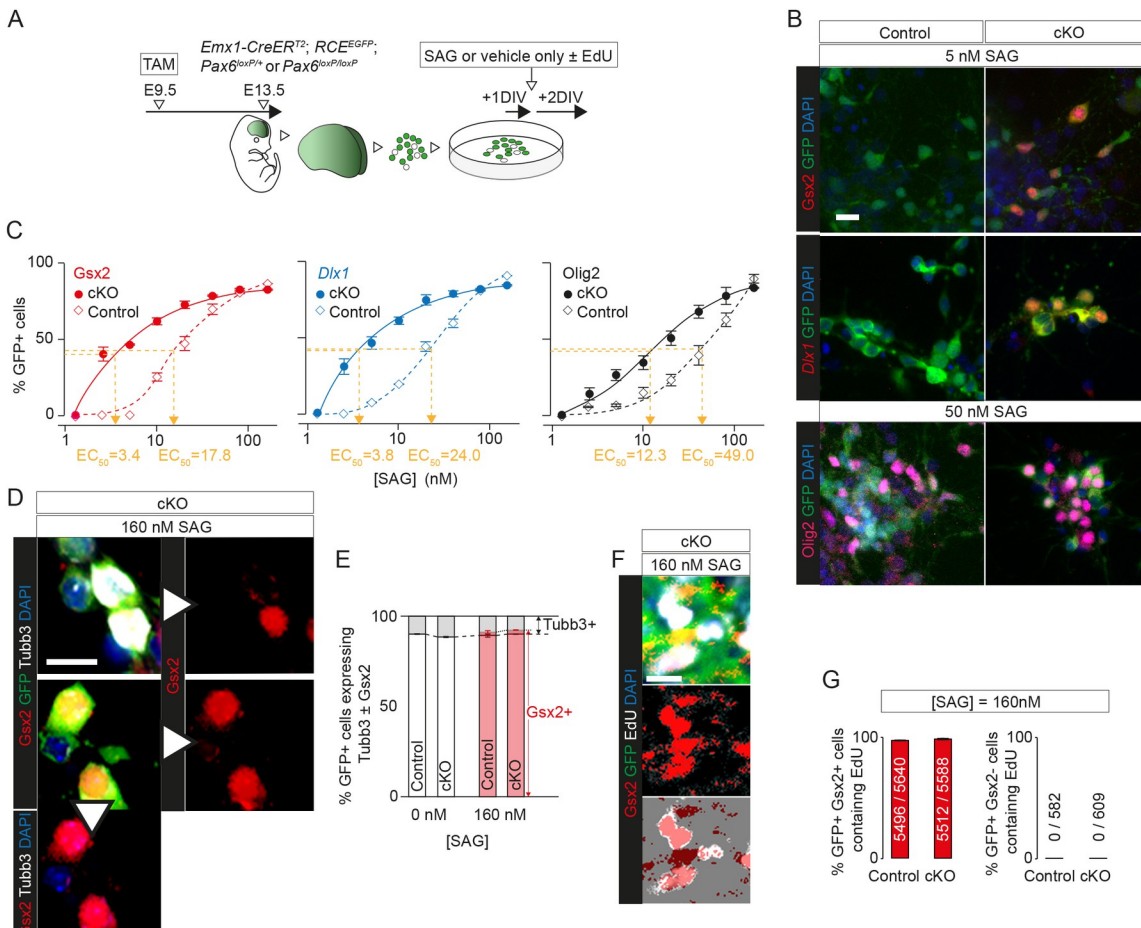

**Fig 8. *Pax6* deletion affected concentration-response of cortical cells to Shh pathway activation. (A)** The experimental procedure for (B-G): TAM was given at E9.5 to generate *Pax6* cKO and control embryos, with Cre activation revealed by GFP expression; E13.5 cortex was dissociated, cells were treated with SAG or vehicle alone, and with EdU in some cases, after 1 DIV, and were analysed after a further 2 DIV. **(B)** Examples of labeling of E13.5 control and *Pax6* cKO cells grown in dissociated culture with 5 nM or 50 nM SAG. Labeling was for DAPI and GFP with Gsx2, *Dlx1*, or Olig2. Scale bar: 0.01 mm. **(C)** Graphs of concentration-responses to SAG (measured as percentages of GFP+ cells expressing Gsx2, *Dlx1*, or Olig2). Data are averages (±SEM; $n$ = 3 independent experiments each), with EC$_{50}$s for each response curve. Two-way analyses of variance were conducted. For Gsx2: significant effects of genotype (f (1,32) = 798.9, $p < 0.001$) and SAG concentration (f(7,32) = 1,138, $p < 0.001$) and significant interaction effect (f(7,32) = 123.5, $p < 0.001$). For *Dlx1*: significant effects of genotype (f(1,32) = 763.6, $p < 0.001$) and SAG concentration (f(7,32) = 1,011, $p < 0.001$) and significant interaction effect (f(7,32) = 91.90, $p < 0.001$). For Olig2: significant effects of genotype (f(1,32) = 177.4, $p < 0.001$) and SAG concentration (f(7,32) = 415.1, $p < 0.001$) and significant interaction effect (f(1,32) = 15.88, $p < 0.001$) (Sheets A-C in S8 Data). **(D)** Examples of labeling of E13.5 control and *Pax6* cKO cells grown in dissociated culture with 160 nM SAG. Labeling was for DAPI, GFP, Gsx2, and Tubb3. Examples include GFP+ cells that were Gsx2+, Tubb3−; Gsx2+, Tubb3+; Gsx2−, Tubb3+. Scale bar: 0.01 mm. **(E)** Average percentages (±SEM; $n$ = 4 independent experiments each) of GFP+ control or *Pax6* cKO cells with or without 160 nM SAG that expressed Tubb3, Gsx2, or both (Sheet D in S8 Data). **(F)** Examples of labeling of E13.5 control and *Pax6* cKO cells grown in dissociated culture with 160 nM SAG. Labeling was for DAPI, GFP, and EdU. Examples include GFP+ cells that were Gsx2+, EdU+; Gsx2+, EdU−. Scale bar: 0.01 mm. **(G)** Average percentages (±SEM; $n$ = 4 independent experiments each) of GFP+ cells that contained EdU among the Gsx2+ and Gsx2− populations in cultures from control and *Pax6* cKO cortex treated with SAG. Total numbers, across all cultures, of GFP+ cells that contained EdU over total numbers of GFP+ cells are stated for each condition (Sheet E in S8 Data). DIV, day in vitro; EdU, 5-ethynyl-2′-deoxyuridine; GFP, green fluorescent protein; *Pax6* cKO, *Pax6* conditional knockout; SAG, signaling agonist; TAM, tamoxifen.

incorporated by most of the GFP+ cells that had activated Gsx2 in response to SAG, but by none of those that remained Gsx2-negative. We concluded that the cells that were the most highly differentiated were the least susceptible to the effects of SAG.

## Bmps contributed to regional differences in ectopic gene activation

We next considered whether Pax6 is also involved in regulating the responses of developing cortical cells to other morphogens. Previous research has shown that the embryonic dorsal telencephalic midline is a rich source of Bmps, including Bmps 4 to 7 [138] and that Bmps can inhibit the expression of genes involved in the specification of GABAergic neurons [139]. Our evidence indicated that many aspects of cortical Bmp signaling remained close to normal in *Pax6* cKO cortex. Our RNAseq data showed that Pax6 removal had no detectable effects on the expression of mRNAs for any of the Bmps and identified only 2 canonical Bmp signaling pathway genes with significantly altered expression levels in *Pax6*-deleted cortex (both only at E13.5: *Bmpr1b* LFC = −0.76 rostrally and −0.64 caudally; *Smad3* LFC = −0.35 rostrally and −0.28 caudally) (S1 Table). Second, phospho-Smad1/5/9 immunoreactivity, whose levels correlate positively with Bmp activity and, therefore, tend to be higher in medial than in lateral embryonic cortex [140–143], showed a similar pattern in control and *Pax6* cKO embryos (S18A Fig).

We tested the effects of Bmp4 on the expression of *Gsx2* and *Prdm13*, whose spatial expression patterns were altered in different ways by *Pax6* deletion. We added increasing doses of Bmp4 to cultured E13.5 control and *Pax6* cKO cortical slices and measured gene expression levels with qRT-PCR and visualized expression patterns in sections (Fig 9A). We found that addition of Bmp4 to *Pax6* cKO cortex lowered overall *Gsx2* mRNA levels and caused loss of Gsx2-expressing cells in sections (Fig 9B and 9C).

The effects of Bmp4 on *Prdm13* expression were more complex (Fig 9D and 9E). Whereas addition of Bmp4 at concentrations <1 µg mL$^{-1}$ had no detectable effect on overall levels of *Prdm13* measured with qRT-PCR, and higher concentrations suppressed overall expression (Fig 9D), in situ hybridizations in sections revealed that addition of Bmp4 actually increased *Prdm13* expression in lateral cortex (Fig 9E, green arrows). A possible reason why this was not reflected in the overall levels of *Prdm13* mRNA was that it appeared to be offset by decreased expression in medial cortex (Fig 9E, asterisk). It seemed possible that *Prdm13* responded biphasically to Bmp activation in *Pax6* cKO cortex, such that (i) *Prdm13* was activated in the range of Bmp activation levels that existed endogenously in medial cortex or were achieved in lateral cortex after exogenous application of Bmp4; and (ii) *Prdm13* expression was suppressed at the relatively higher Bmp activation levels that were achieved medially when endogenous Bmp activation was supplemented by exogenous Bmp4. Our findings indicated that cells in both medial and lateral *Pax6* cKO cortex were competent to express *Prdm13* and whether they did so depended on them receiving the requisite signals.

Fig 9F outlines how Shh and Bmp4 might combine to generate the spatial expression patterns of *Gsx2* and *Prdm13* in *Pax6* cKOs (Fig 3). In this model, *Gsx2* activation in cells exposed to suprathreshold levels of Shh is counteracted in medial cortex by relatively high levels of Bmp signaling, preventing Gsx2 expression in this region. *Prdm13* is activated by intermediate levels of Bmp signaling but suppressed by the highest levels, which might explain the lowering of *Prdm13* expression very close to the dorsal midline (Fig 3F and 3G).

We found that Bmp4 did not suppress the expression of *Olig2* in *Pax6* cKO cortex (S18B Fig). The selective effect of Bmps on *Gsx2* expression with no effect on *Olig2* expression suggested one possible reason why, in vivo, Gsx2-expressing cells were initially less widespread than Olig2-expressing cells following Pax6 deletion (S9A, S9B, and S9H Fig).

## Conclusions

Fig 9G illustrates our main findings using Waddington's epigenetic landscape in which the developmental trajectory of a cell is represented as a ball rolling downhill through valleys

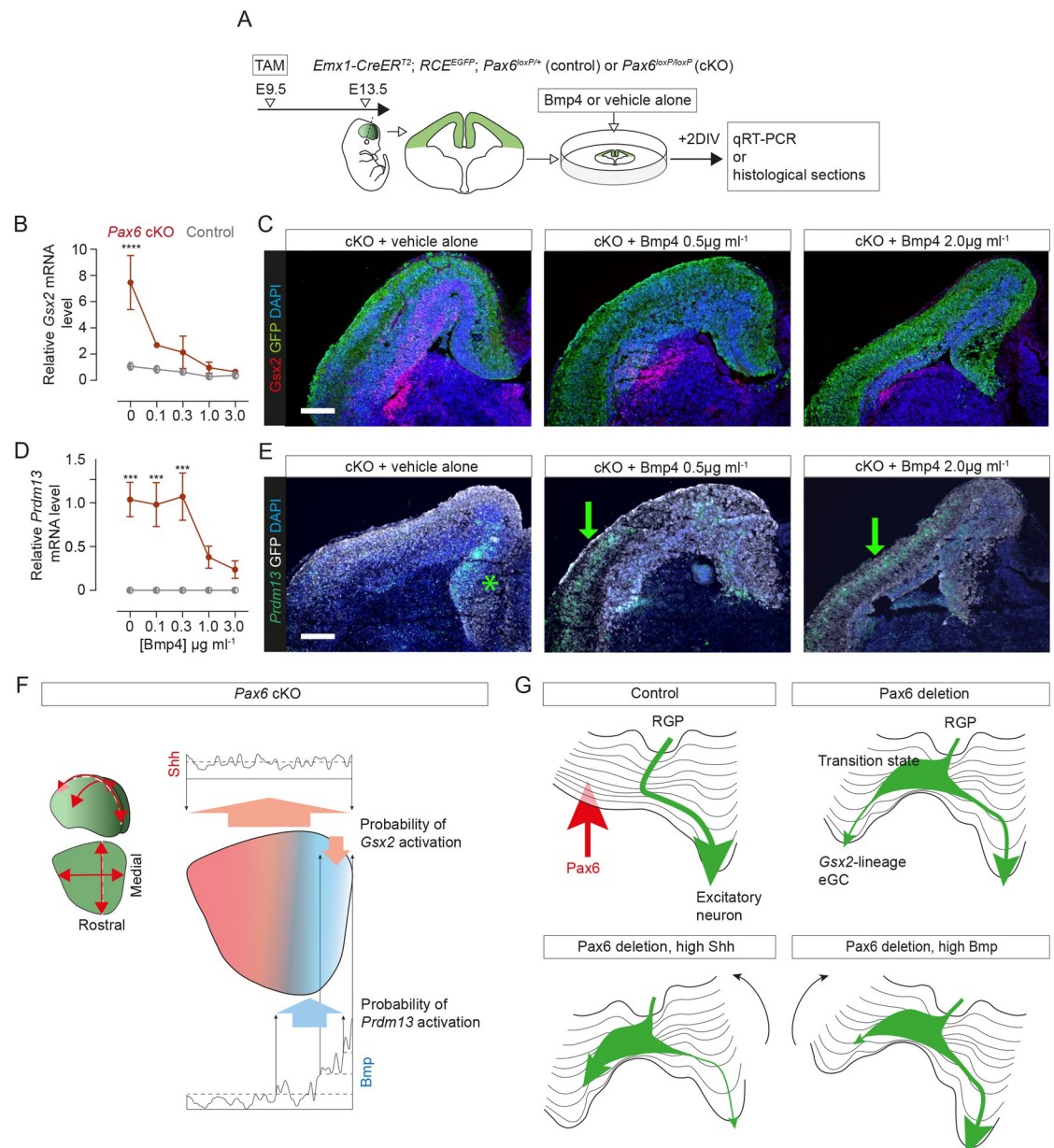

**Fig 9. Morphogen regulation of ectopic gene expression following Pax6 loss. (A)** The experimental procedure for (B-E): TAM was administered at E9.5 to delete either one (control) or both (cKO) *Pax6* allele(s), with Cre-deleted cells expressing GFP; coronal slices were cultured on E13.5 with Bmp4 or vehicle alone for 2 DIV; slices were analysed using qRT-PCR or sectioned. **(B)** Concentration-response measured using qRT-PCR: *Gsx2* levels (averages ± SEM; values were relative to the average level in control cortex treated with 0 Bmp4) in control and *Pax6* cKO slices with increasing concentrations of Bmp4 ($n = 3$ independent cultures at each concentration). Two-way ANOVA showed significant effects of genotype ($p < 0.001$), of Bmp4 concentration ($p < 0.005$), and an interaction effect ($p < 0.01$). Differences between genotypes at each Bmp4 concentration were tested with Bonferroni's method for comparison of means (****, $p < 0.001$) (S9 Data). **(C)** Immunoreactivity for Gsx2 and GFP in telencephalic slices from *Pax6* cKOs cultured with vehicle alone or Bmp4. Scale bar: 0.1 mm. **(D)** Concentration-response measured using qRT-PCR: *Prdm13* levels (averages ± SEM; values are relative to the average level in *Pax6* cKO cortex treated with 0 Bmp4) in control and *Pax6* cKO slices with increasing concentrations of Bmp4 ($n = 3$ independent cultures at each concentration). Two-way ANOVA showed significant effects of genotype ($p < 0.005$), of Bmp4 concentration ($p < 0.05$), and an interaction effect ($p < 0.05$). Differences between genotypes at each Bmp4 concentration were tested with Bonferroni's method for comparison of means (***, $p < 0.005$) (S9 Data). **(E)** In situ hybridizations for *Prdm13* and immunoreactivity for GFP in telencephalic slices from *Pax6* cKOs cultured with vehicle alone or Bmp4. Green arrows indicate *Prdm13* expression in lateral cortex. Scale bar: 0.1 mm. **(F)** A hypothesis of how Shh and Bmp4 might combine to generate the observed spatial patterns of *Gsx2* and *Prdm13* expression in the embryonic cortex after Pax6 deletion. Deletion might increase the probability of *Gsx2* being activated in cells exposed to physiological levels of Shh above a

threshold (broken line). In the medial cortex, exposure to the levels of Bmp above a threshold (central broken line) might reduce the probability of Gsx2 activation. Cells exposed to intermediate levels of Bmp (between upper and lower broken lines) might have an increased probability of expressing *Prdm13*. (**G**) Waddington's epigenetic landscape analogy, used to illustrate our main conclusions. A saddle-node bifurcation illustrates Pax6's normal action, closing a valley on the left (RGP). Pax6 deletion opens this valley, creating a subcritical pitchfork bifurcation where cells emerging from the transition state can enter either of 2 valleys (eGC). Increasing exposure to Shh tilts the landscape to the left making it more likely that the cell will enter the open valley on the left; increasing exposure to Bmp has the opposite effect. DIV, day in vitro; eGC, ectopic GABAergic cell; GFP, green fluorescent protein; *Pax6* cKO, *Pax6* conditional knockout; qRT-PCR, quantitative real-time PCR; RGP, radial glial progenitor; TAM, tamoxifen.

created by the actions of that cell's genes [144–146]. Pax6's actions ensure that, under physiological conditions, cortex-born cells progress unerringly toward their excitatory neuronal fate. If Pax6 is deleted, alternative trajectories become available (at what are known as pitchfork bifurcations; [146,147]). Morphogens such as Shh and Bmps, whose concentrations vary dynamically and regionally, have the effect of tilting each cell's landscape to a variable degree, thereby influencing which alternative is likely to be adopted [145]. Pitchfork bifurcations are associated with unstable and reversible transition states in which the cell shows a mixed identity between the original and the destination states [146]. We envisage that the aPs we identified by scRNAseq in *Pax6* cKO cortex are examples of cells in this state.

## Discussion

### Pax6 affects the competence of cortical cells to respond to signaling molecules

Regionally expressed transcription factors such as Pax6 can contribute to the development of cellular diversity in several ways: (i) by regulating cells' competence to respond to signaling molecules around them; (ii) by controlling their production of intercellular signaling molecules; and (iii) by acting internally to influence cellular development without affecting intercellular signaling. Our present study has highlighted the overriding importance of Pax6 in the first of these mechanisms during cortical neurogenesis, although it is probably involved to some extent in all three.

Pax6 is expressed in many cell types in structures including the eye, brain, spinal cord, and pancreas [148,149]. Its regulation of cellular competence during cerebral cortical neurogenesis is mirrored by some of its actions in other tissues. For example, Pax6 is involved in establishing the competence of different sets of murine thalamic cells to produce either glutamatergic or GABAergic neurons [150]; PAX6 restricts the competence of neuroepithelium derived from human embryonic stem cells to express markers of GABAergic progenitors in response to SHH application [151]; Pax6 regulates the competence of cells to respond to inductive signals during vertebrate and invertebrate eye development [152,153]. It is important to stress, however, that the nature of Pax6's effects vary considerably in different contexts: for example, whereas it limits cells' competence to express genes such as *Gsx2* during cerebral corticogenesis, it has an opposite effect on *Gsx2* expression in diencephalic cells [36,154]. This and other context-dependent differences likely arise at least in part from differences in its combinatorial expression with other transcription factors, such as Foxg1 in the telencephalon [36] or Irx3 in the diencephalon [150].

Regarding the second possibility listed above—that Pax6 regulates the production of intercellular signaling molecules that affect how other cells develop—our RNAseq data identified very few changes in the expression of genes encoding signaling molecules. These did, however, include a change in *Ffg15* expression, which was up-regulated in rostral cortex following *Pax6* deletion. This is potentially interesting because experimentally induced up-regulation of *Ffg15* in embryonic cortex can facilitate the activation of ventral telencephalic genes [155]. Although previous work on the embryonic cortex of mouse chimeras containing mixtures of wild-type

and *Pax6*-null cells found no evidence that wild-type cells were affected by abnormal signaling from *Pax6*-null cells [23], it remains possible that altered signal production by *Pax6*-deleted cells affects other *Pax6*-deleted cells, due to their altered competence.

Regarding the third possibility listed above, Pax6's regulation of genes not directly involved in intercellular signaling almost certainly contributes to some of its cellular actions. For example, we have argued previously that Pax6 limits cortical progenitor cell cycle rates through mechanisms that include direct repression of *Cdk6* [35]. Nevertheless, such explanations are probably incomplete. For example, Shh is a mitogen [156–158] and the highly proliferative nature of many eGCs identified in the present study might be caused in part by cortical cells' having enhanced responses to Shh that include effects on their proliferation.

Our overall conclusion is that, while Pax6's dominant role during cortical neurogenesis is to limit the developmental potential of cortical cells to respond to signaling molecules that are carrying out other functions around them, it is very likely to act in other ways too.

## Pax6 raises cortical cellular thresholds for responses to morphogens such as Shh

In developing multicellular organisms, the reproducible production of distinct specialized cell types in specific locations involves cells acquiring information about their relative positions through interactions with their extracellular environment. For example, one iconic model proposes that cells in a contiguous population, or field, acquire positional information from their levels of exposure to substances distributed in concentration gradients across the field [159]. It is thought that such mechanisms specify the 2 major domains of the rostral neural plate, a ventral domain producing the basal ganglia and a dorsal domain producing the cerebral cortex. The morphogen Shh is one example of a substance whose concentration varies dorsoventrally as these 2 domains emerge, with its high ventral levels contributing to the specification of ventral fates [160,161]. As the rostral neural tube expands and complexifies, however, so do the distributions of morphogens that affect its subsequent development. For example, the closure of the neural tube to create the nervous system's ventricular system gives numerous morphogens, including Shh, widespread access to developing telencephalic cells via the CSF [117] and interneurons migrating into the cortex produce numerous morphogens, including Shh [114–116,162]. Previous work has suggested that Shh levels in embryonic cortex are high enough that they could potentially induce the ectopic expression of ventral marker genes [155]. These observations bring into sharp focus the importance of mechanisms regulating not only the distributions of morphogenetic substances but also the responses of cells to them.

A classic proposition is that cells have intrinsic thresholds determining how aspects of their development are influenced by surrounding morphogens [159]. Our work suggests that Pax6 raises some of these intrinsic thresholds in embryonic cortical cells, preventing them from activating erroneous programs of gene expression in response to physiological levels of Shh, Bmps, and possibly other morphogens around them [89,114–116,155,163–167]. The reasons for thinking in terms of threshold adjustment is that Pax6 does not produce an absolute block under all circumstances to the possibility of morphogen-induced ectopic gene expression. Previous work has shown that nonphysiological elevation of Shh signaling in Pax6-expressing embryonic cortex can activate the ectopic expression of ventral telencephalic marker genes [39,133–137]. Our in vitro data showed that Shh can induce Pax6-expressing cortical cells to express ventral telencephalic markers in a concentration-dependent manner, but *Pax6* cKO cells respond with much greater sensitivity.

Pax6 acts selectively on the various cellular process affected by morphogen signaling. It does not prevent morphogens such as Shh and Bmps having important physiological functions

in processes including cortical cell proliferation, migration, and neuronal morphogenesis [114,115,166,168,169]. This selectivity might be a consequence of Pax6 acting around the point where the intracellular signal transduction pathways target the genome; for example, it might reduce the ability of molecules acting late in these pathways, such as Gli and Smad transcription factors [158,161,166,170], to bind some of their potential genomic target sites. At least some such effects might involve indirect actions, via effects of Pax6 on the expression of other transcription factors. For example, Ascl1 is a possible intermediate. We found that its up-regulation in *Pax6* cKO cortex is widespread and is associated with multiple lineages. Previous work has shown (i) that forced expression of Ascl1 in embryonic cortex activates the ectopic expression of ventral markers such as *Dlx1* and *Gad1* in some cortical cells [40]; (ii) that Ascl1 promotes chromatin accessibility during neurogenesis; and (iii) that *Olig2* is one of Ascl1's direct targets [171].

A remarkable feature of the mechanisms that regulate the development of multicellular organisms is the extent to which the same morphogens are reused in multiple diverse mechanisms as the embryo grows. It seems highly likely that adjustments to how cells at different locations and ages respond to the same morphogens would have been instrumental in allowing the diversification of morphogen function during evolution. The evolution of mechanisms limiting cellular competence, such as those described here, would have allowed a morphogen to acquire new functions by minimizing the risk of the beneficial consequences being offset by its preexisting potential to induce changes that would be undesirable in the new context.

## Pax6's effects on Shh signaling might control the timing of cortical excitatory neuron specification

Previous studies have shown that Shh can repress *Pax6* expression in diverse tissues including the telencephalon [134,150,155,172,173,174]. The fact that Shh levels are highest in ventral telencephalon from the earliest stages of its development is likely to be a major factor establishing the ventral-low versus dorsal-high difference in *Pax6*'s expression levels [175–178]. Our findings indicate that, as dorsal telencephalic development progresses, its high levels of Pax6 repress the potential actions of morphogens including Shh itself, thereby opening a time window for the unhindered production of a normal complement of cortical excitatory neurons.

When this time window opens, Pax6 is normally expressed by a large proportion of cortical cells, with highest levels in RGPs [11,62,179]. For example, our scRNAseq data showed that more than 70% of cells express *Pax6* at E13.5. As the cortex ages, a progressively smaller proportion of cells express *Pax6* (approximately 50% at E14.5 in our scRNAseq data) as non-RGP populations, including non-RGPs such as IPs, expand. This decline coincides with the normal cortical activation from approximately E16.5 onward of *Gsx2* and *Olig2* in a small proportion of SVZ cells [89]. These cells are thought to be tri-potential IPs that generate interneurons for the olfactory bulb, cortical oligodendrocytes, and astrocytes. Their generation requires Shh, and they and their daughters express many of the same genes as those activated earlier in Pax6-deleted cortex. It seems plausible that, in normal corticogenesis, the waning influence of Pax6 in late progenitor populations allows the time window for the focused production of cortical excitatory neurons to close.

## Pax6 loss generates cortical cells with varying degrees of similarity to normal cell types

Previous studies have shown that even if cortical progenitors lose the ability to produce functional Pax6, whether due to constitutive or conditional mutation, they can still generate many cells that migrate into the CP to form layers containing neurons with excitatory morphologies

and connections resembling those in normal cortex [11,78]. In the present study, we confirmed and extended these previous conclusions: The transcriptional profiles of *Pax6*-deleted cells that were differentiating into excitatory neurons and the electrophysiological properties of *Pax6*-deleted CP neurons were indistinguishable from those of control cells. Overall, we found no evidence that Pax6 is required to instruct the specification of cortical excitatory neurons from cortical progenitors.

Our analyses indicated that some populations of cells that are present in normal cortex expanded following Pax6 loss. This applied to cells in the dorsomedial part of the embryonic cortex. The expansion of these populations might have been due, at least in part, to an increased sensitivity to Bmps, which are present at higher levels dorsomedially and are involved in the specification of dorsomedial cell identities during normal development [138,139,140,142,180].

Previous reports have suggested that *Pax6*-deleted cortical progenitors generate cells with a dLGE identity that go on to produce immature olfactory bulb interneurons, as do dLGE cells in normal telencephalon [21,24,181]. While some evidence supports this interpretation, we suggest that it should be treated with caution. Most tellingly, we found that many of these cells do not respond to Gsx2 loss in the same way as normal dLGE cells. This difference might reflect intrinsic differences with normal dLGE cells and/or differences in their extracellular environments. Similarly, intrinsic and/or environmental factors might contribute to the accumulation of abnormal cell types beneath the CP and their subsequent death. Overall, it seems unwise at present to assume that these abnormal cell types are necessarily representative of cell types present in normal telencephalon.

## Methods

### Animals

All experimental procedures involving mice were regulated by the University of Edinburgh Animal Welfare and Ethical Review Body in accordance with the UK Animals (Scientific Procedures) Act 1986 (licensing number P53864D41). All the alleles used and their genotyping have been described before: *Btg2-GFP* [97]; *CAGG-Cre^ERTM* [182]; *Emx1-Cre* [183], *Emx1--Cre^ERT2* [33], *Foxg1^loxP* [184], *Gsx2-Cre* [33], *Pax6^loxP* [185], *Pax6^Sey* [186], *Gsx2^loxP* [47]; *RCE: loxP* (*R26R CAG-boosted enhanced green fluorescence protein (EGFP)* Cre reporter) [37]; *Shh⁻* [187]. Activation of CreERT2 or CreERTM was achieved by giving pregnant females 10 mg of tamoxifen by oral gavage.

### Quantitative real-time polymerase chain reaction (qRT-PCR)

Total RNA was extracted using either the RNeasy Plus Mini kit (Qiagen) for samples taken directly from cortex or the RNeasy Micro kit (Qiagen) for cultured cortical slices, and cDNA was synthesized with a Superscript reverse transcriptase reaction (Thermo Fisher Scientific).

*Pax6*: qRT-PCR was performed using a DNA Engine Opticon Continuous Fluorescence Detector (MJ Research) and a Quantitect SYBR Green PCR kit (Qiagen). We used the following primer pairs. *Pax6*: 5′-TATTACGAGACTGGCTCCAT-3′ and 5′-TTGATGACACACTG GGTATG-3′; *Gapdh*: 5′- GGGTGTGAACCACGAGAAAT-3′ and 5′-CCTTCCACAATGC CAAAGTT-3′. We calculated the relative abundances of *Pax6* and *Gapdh* transcripts for each sample. For each biological replicate, we ran 3 technical replicates.

*Crabp1*, *Gsx2*, *Olig2*, *Prdm13*: qRT-PCR was performed using the Applied Biosystems StepOnePlus RT-PCR machine (Thermo Fisher Scientific) and a TaqMan Gene Expression Assay (Thermo Fisher Scientific) containing a TaqMan probe with a fluorescein amidite dye label on the 5′ end and a minor groove binder and nonfluorescent quencher on the 3′ end. TaqMan

probes used in this experiment, designed and supplied by Thermo Fisher Scientific, were as follows. *Crabp1*: Mm00442775_g1; *Gsx2*: Mm00446650_m1; *Olig2*: Mm01210556_m1; *Prdm13*: Mm01220811_m1. For each sample, we ran 3 technical replicates with no-template control. Target gene expression was calculated as fold change relative to gene expression in the vehicle alone condition.

## Western blots

Proteins extracted from homogenized and lysed cortical tissue were resolved on denaturing gels. Primary antibodies against Pax6 (1:500; rabbit; Covance) and β-actin (1:2,000; rabbit; Abcam) were detected with Alexa-coupled secondary antibodies, and blots were quantified using the LI-COR scanning system (LI-COR Biosciences). The intensity of each Pax6 band was normalized to that of its corresponding β-actin band.

## Immunostaining and in situ hybridization on tissue sections

Pregnant mice were killed by cervical dislocation, and embryonic brains were dissected into 4% paraformaldehyde (PFA); postnatal mice were perfused transcardially with 4% PFA; cultured cortical slices were placed in 4% PFA. Samples were fixed overnight at 4°C, and cryosections were cut usually at 10 μm or at 2 μm in some cases.

**Immunostaining.** Primary antibodies used in this study were as follows. Anti-Ascl1 (mouse; BD Biosciences); anti-Bromodeoxyuridine (BrdU) (mouse; clone B44, BD Biosciences); anti-Caspase 3 (rabbit; Millipore); anti-Crabp1 (rabbit; Cell Signaling); anti-Eomes (rabbit; Abcam); anti-Foxg1 (mouse; kindly provided by Steven Pollard, University of Edinburgh, UK; [188]); anti-GFP (rabbit or goat; Abcam); anti-Gsx2 (rabbit; Merck); anti-Histone H3 (phospho S10) (rabbit; Abcam); anti-mCherry (chicken; Abcam); anti-Olig2 (rabbit; Millipore); anti-Pax6 (mouse; clone AD2.38, described in [189]); anti-pSmad1/5/9 (rabbit; Cell Signaling); anti-Shh (rabbit; kindly provided by Genentech); anti-Slc17a7 (formerly Vglut1) (rabbit; Synaptic Systems); anti-Sox9 (rabbit; Millipore); anti-Turbo GFP (mouse; Origene).

For fluorescence immunostaining, sections were incubated with secondary antibodies (1:200) coupled with Alexa Fluor 488 (Abcam) or 568 (Abcam) or Cy3 (Abcam) and then incubated with diamidino-2-phenylindole (DAPI; 1:1,000; Thermo Fisher Scientific) before being mounted with Vectashield HardSet (Vector Laboratories) or ProLon Gold Antifade Mountant (Thermo Fisher Scientific).

For colorimetric immunostaining, sections were incubated with biotinylated secondary antibodies followed by Avidin Biotin Complex (ABC kit, Vector Laboratories) and then diaminobenzidene (DAB, Vector Laboratories). Sections were mounted in DPX (Sigma).

**In situ hybridization.** Riboprobes used in this study were as follows. *Ascl1* and *Gad1* (kindly provided by Francois Guillemot, Francis Crick Institute, UK); *Dlx1* (kindly provided by Yorick Gitton, INSERM Institut de la Vision, France); *Foxg1* (kindly provided by Vassiliki Fotaki, University of Edinburgh, UK); *Gsx2* (kindly provided by Kenneth Campbell, Cincinnati Children's Hospital, USA); *Gsx1* and *Neurog2* (kindly provided by Thomas Theil, University of Edinburgh, UK); *Prdm13* (kindly provided by Tomomi Shimogori, Riken Centre for Brain Science, Japan). Templates for *Robo3*, *Rlpb1*, *Pde1b*, *Igsf11*, *Heg1*, and *Neurod4* riboprobes were synthesized from mouse embryonic brain cDNA using the following PCR primers (convention: gene, forward primer, reverse primer). *Heg1*, 5′-ACTTCCAAATGTCCCCATA CAC-3′, 5'-CCAGCCCAATCTATTAAAGTGC-3'; *Igsf11*, 5′-TCAGTGCCCTCTCTTCCG-3′, 5'-CAGGCCACTTCACACACG-3'; *Neurod4*, 5′-TGGAATGCTCGGAACCTTAA-3′, 5′-TACAGGAACATCATAGCGGG-3′; *Pde1B*, 5′-GCTGACTGATGTGGCAGAAA-3′, 5′-AGAATCCCAATGGCTCCTCT-3′; *Rlbp1*, 5′-TTCCTCCTGCGCTTCATC-3′, 5′-TTGG

GATGAGGTGCCACT-3′; *Robo3*, 5′-GCTGTCCTCCGTGATGATTT-3′, 5′-AAATTGTGGT
GGGACGTGAA-3′.

Riboprobes were labeled with digoxigenin (DIG) or dinitrophenol (DNP). DIG-labeled
probes were synthesized using a DIG RNA labeling mix (Roche). DNP-labeled probed were
synthesized using a similar process but with the DIG labeling mix replaced by a 20× NTP stock
solution (20 mM each ATP, GTP, CTP, 13 mM UTP, Thermo Fisher Scientific) and a 20×
DNP-11UTP stock solution (7 mM; Perkin Elmer). For fluorescence in situ hybridisation,
probes were detected using an anti-DIG peroxidase (Roche) or anti-DNP peroxidase (Tyra-
mide Signal Amplification (TSA) Plus DNP (HRP) kit, Akoya Bioscience) followed by TSA-
plus cyanine 3 or TSA-plus fluorescein (Akoya Bioscience). When required, amplification was
achieved using TSA Biotin (Akoya Bioscience). For fluorescence double in situ hybridization,
the probes were detected sequentially, and the slides were incubated in 10 mM HCl before
detection of the second probe. When fluorescence in situ hybridisation was followed by immu-
nofluorescence, a microwave antigen retrieval step (20 min in 10 mM sodium citrate) was
required at the end of the in situ hybridisation protocol prior to proceeding with
immunostaining.

## Bulk RNAseq analysis of *Pax6* cKO, *Pax6 Gsx2* double cKO, and control cortex

Pregnant mice were killed by cervical dislocation following isoflurane overdose and embryos
were removed. Quality control prior to RNAseq involved the use of immunohistochemistry on
one cerebral hemisphere from every *Pax6* cKO or *Pax6 Gsx2* dcKO embryo and from all con-
trol littermates to check for efficient deletion of Pax6 or Pax6 and Gsx2. The other hemisphere
from brains that passed this quality control was processed for RNAseq as follows. The EGFP
reporter was used to guide the dissection to ensure only EGFP+ cortex was included. The ros-
tral and caudal halves of each cortex were separated and frozen on dry ice. Cortical pieces
from 3 E13.5 embryos or 4 E12.5 embryos were pooled to produce each sample. Total RNA
was extracted from each pool using an RNeasy+ Micro kit (Qiagen). Poly-A mRNA was puri-
fied and TruSeq RNA-Seq libraries (Illumina) were prepared and sequenced (100 base paired-
end; Illumina, HiSeq v3).

Sequencing gave an average of 108 million reads per sample. Following adapter trimming
and standard quality checks, reads were aligned using STAR v. 2.4.0a to GRCm38.p3 (mm10)
genome build obtained from Ensembl release 77. Reads were counted using featureCounts v.
1.4.5-p from the Subread package, and only fragments with both reads properly aligned to
exon regions were counted.

DESeq2 [190] and edgeR [191] were used to identify genes that were differentially expressed
(DE) between conditions. When using DESeq2, package functions were used with default
parameters and DE genes were defined as those with adjusted $p$-value $\leq 0.05$ after Benjamini–
Hochberg correction for multiple testing. When using edgeR, sample dispersion was estimated
using default parameters and pairwise exact tests were used to compare expression under dif-
ferent conditions: DE genes were defined as those with false discovery rate (FDR) $\leq 0.05$. We
included all genes identified by either of these packages for further consideration as DE genes
(most DE genes were identified by both packages).

## Single-cell RNAseq (scRNAseq) analysis of *Pax6* cKO and control cortex

Pregnant mice were killed by cervical dislocation following isoflurane overdose and embryos
were removed. Embryos were placed in ice-cold Earle's Balanced Salt Solution (EBSS) and
examined under a fluorescence stereomicroscope for the presence of EGFP. EGFP-positive

brains were used for the scRNAseq experiment; EGFP-negative brains from littermates were used as negative controls for EGFP signal calibration in subsequent FACS. The rostral half of the cortex from one hemisphere of each GFP-positive brain was isolated and used for tissue dissociation; the other hemisphere was fixed with 4% PFA and used in immunohistochemistry to check for efficient deletion of Pax6. The tails from the embryos were collected for genotyping to confirm *Pax6^flox^* copy number.

Tissue was dissociated in 30 U/mL papain reagent containing 0.4% DNase (Worthington) for 30 min at 37˚C. Samples were gently triturated using glass pipettes (BrainBits LLC) and sieved through a 40-µm cell strainer (pluriSelect) to remove cell clumps. Dissociated cells were resuspended in basic sorting medium containing 2% heat-inactivated fetal bovine serum. Trypan blue was added to sample aliquots to allow quantification of cell density and viability. EGFP-positive cortical cells were selected using the FACSAria II (BD Biosciences). Cells were stained with DAPI to select against cell debris, cell doublets, and dead cells. The system was then calibrated with EGFP-negative cells, before selecting approximately 20,000 EGFP-positive cells for each sample. FACSDiva 8.0.1 software (BD Biosciences) was used to process all flow cytometry data.

Three embryos from a pregnant mouse were used to prepare each E13.5 library. For E14.5 libraries, *Pax6* cKO embryos from 3 pregnant mice and control embryos from 2 pregnant mice were pooled together. All libraries from E13.5 and E14.5 were produced from male embryos; none of the cells across all 4 libraries expressed *Xist*.

The 10x Genomics Chromium Controller and Single Cell 3′ Reagent Kits (v.2 for E13.5; v.3 for E14.5) were used for library construction, as per manufacturer's instructions. Cell density and viability were quantified using an automated cell counter as part of the quality control before loading onto the controller chip. Each sample was its own library. For each library, 3,000 to 4,000 cells were targeted for capture. Each library was constructed with a specific sample barcode and standard Illumina paired-end constructs. Two libraries from each embryonic age were sequenced simultaneously using the Illumina NovaSeq 6000 S1 flow cell. Raw sequencing reads were processed through the CellRanger pipeline (v.2 at E13.5; v.3 at E14.5; 10x Genomics) to produce a gene-cell count matrix using the mm10 mouse genome as reference. Approximately 60,000 to 100,000 reads per cell were generated to achieve optimal sequencing depth.

All data were subjected to SoupX [192] to remove ambient RNA captured in droplets. To filter out cells giving low-quality data, cutoffs on unique molecular identifier (UMI), gene count, and mitochondrial gene percentage were applied. Cells with gene counts >3 standard deviations from the mean were excluded: Depending on sequencing depth, cells expressing <approximately 1,000 and >approximately 5,500 genes were typically excluded. Cells expressing extremely high numbers of mitochondrial genes (>5% at E13.5; >10% at E14.5) were also excluded.

Clustering analysis was performed using the Seurat v.3.0 bioinformatics pipeline (https://github.com/satijalab/seurat). For each dataset, a Seurat object was created. UMI counts were natural log-normalized using log1p with scale factor of 10,000 using the NormalizeData function. Individual gene expression on UMAP was constructed with natural log-normalized expression using the FeaturePlot function. Expression values for each gene across all cells was standardized using z-score transformation implemented in the ScaleData function. A cell cycle phase was assigned to each cell using the Seurat pipeline. The datasets were not regressed with cell cycle scores as this might have hindered identification of the physiological relevance of cell cycle in the temporal progression of cortical neurogenesis. For integrated analysis across the control and *Pax6* cKO datasets, normalization and feature selection for highly variable genes in each library were performed independently. Based on the selected features, the control and

*Pax6* cKO datasets were passed through the FindIntegrationAnchors function to identify the cross-dataset anchors for data integration. The resulting integration anchors were then implemented in the IntegratedData function, producing an integrated Seurat object. Next, the most variable genes in the integrated dataset were identified and used for PCA. The most statistically significant principal components were used as inputs for nonlinear dimensionality reduction using UMAP implemented in the FindCluster function at a resolution between 0.8 and 1.6 to determine cellular clusters based on k-nearest neighbors and shared nearest neighbor (SNN) graphs. Top cluster marker genes were identified, and DEA across 2 different cellular clusters was carried out using Model-based Analysis of Single-cell Transcriptomics (MAST) implemented in the FindAllMarkers function.

Cell lineage or pseudotime inference was carried out using the Slingshot algorithm (https://github.com/kstreet13/slingshot) [91]. A matrix with reduced dimensionality and cell clustering assignments was taken as input by Slingshot. With the RGP as the initial cell state (or root node), lineage trajectories and branch points were inferred by connecting the cluster medoids with a minimum spanning tree. Patterns of gene expression along pseudotime were computed using the tradeSeq algorithm (htts://statomics.gitbub.io/tradeSeq/index.html) [93]. Gene expression along pseudotime was normalized using log1p.

RNA velocity estimation was generated using the Velocyto (https://github.com/velocyto-team) and scVelo (https://github.com/theislab/scvelo) software packages. Count matrices with cells that had passed quality control were used as inputs for Velocyto. All velocity embeddings were estimated using the stochastic model. For visualization purposes, the RNA velocity embeddings were visualized using the UMAP coordinates produced from the previous cell clustering analysis in Seurat.

## BrdU and ethynyldeoxyuridine (EdU) labeling

Pregnant females were given a single intraperitoneal dose of BrdU or EdU (10 mg mL$^{-1}$, Thermo Fisher Scientific). Subsequent immunohistochemistry was as described above. Cells labeled with EdU were visualized using a Click-iT EdU Alexa Fluor 647 Imaging Kit (Thermo Fisher Scientific).

## Whole-cell electrophysiology

Postnatal mice were anesthetized with isoflurane and killed by decapitation, the brain was quickly removed into artificial cerebrospinal fluid (aCSF) (80 mM NaCl, 2.5 mM KCl, 1.25 mM NaH$_2$PO$_2$, 25 mM NaHCO$_3$, 10 mM glucose, 90 mM sucrose, 0.5 mM CaCl$_2$, 4.5 mM MgSO$_4$), and 400 μm coronal slices were prepared in ice-cold aCSF solution using a vibrating microtome (VT1200S, Leica, Germany). Slices were transferred to aCSF solution for 30 min at 35°C, and current clamp whole-cell recordings were performed using a Multiclamp 700A amplifier (Molecular Devices, Palo Alto, USA). Signals were filtered online at 10 kHz and digitized at 40 kHz and traces stored on the computer using Signal 2 software (Cambridge Electronic Design, UK). Borosilicate glass electrodes (Harvard Apparatus, UK) with a resistance of 3.5 to 7.5 MΩ (Digitimer Research Instruments, UK) were filled with the following solution: 135 mM KMeSO$_4$, 8 mM NaCl, 10 mM HEPES, 0.5 mM EGTA, 0.5 mM Na-GTP, and 4 mM Mg-ATP, 0.5 mM EGTA (pH 7.3), osmolarity adjusted to 285 mOsm. The chamber was perfused at a rate of 3 to 4 mL/min with carbonated (95% O2, 5% CO2) external solution containing the following: 130 mM NaCl, 2.5 mM KCl, 1.25 mM NaH$_2$PO$_4$, 25 mM NaHCO$_3$, 10 mM glucose, 2.5 mM CaCl$_2$, 1.5 mM MgSO$_4$ at 32 to 35°C. Access resistance was monitored online throughout the recording, and if it dropped >20%, the recording was discarded and no further analysis was carried out. In current clamp, small incremental voltage steps (10/25 pA) were

applied for 500 ms to measure cell intrinsic properties. Membrane potential was held at −70 mV. Feature extraction was done blind to genotype using custom scripts written in Signal 2. Some raw.csf recordings of cortical cells were lost and are no longer available.

AP features were extracted from single APs at rheobase. AP threshold was defined as the time at membrane potential at which the slope first exceeded 30 V/s. AP amplitude was defined as height from AP threshold to peak. AHP amplitude (mV) was defined as potential difference between AP threshold and AHP trough. AP latency was defined as the time between start of current step and first AP peak. Properties of trains of APs were determined in response to a current step double rheobase. First-second adaptation ratio and first-last adaptation ratio were calculated by dividing instantaneous frequency of the second/last AP by the first AP instantaneous frequency (this measures the interval between spikes). Spikelet amplitude (mV) was defined as the difference between peak voltage and steady-state voltage during the current step. Input resistance and membrane time constant (tau) were measured using the average of 20 membrane potential responses to negative current steps (−10 or −25 pA) lasting 500 ms. Membrane time constant (tau) was calculated by fitting a single exponential (10% to 90%) to the membrane potential response curve. Capacitance was then calculated using tau/input resistance = capacitance. To test the effects of TTX on spikelets, we used spikelet-inducing current steps (square pulses 0.5 s long; 60 sweeps over 60 s) while the chamber was perfused with external solution either containing TTX (300 nM) or without TTX, to ensure there was no spikelet wane with persistent stimulation. To show TTX blockade of spikelets, which occurred over 30 sweeps, we compared the average amplitudes of spikelets over the first and last 10 sweeps.

Data were analysed statistically and by unsupervised hierarchical clustering. For pairwise comparisons, N's were animals. Data were analysed in R (packages: cluster, ggpubr, dendextend, Nbclust, PairedData, and tidyverse). Preanalysis routines were carried out to inspect data normality by plotting residual plots (qqplots). If data were not normally distributed, a nonparametric test was used. Data for clustering were scaled using base R scale function. For unsupervised clustering, the agglomerative Ward's linkage method was used after calculating Euclidean distances to minimize the variance within clusters [193]. Data features used for clustering are as follows: AHP amplitude, AP halfwidth, AP threshold, input resistance, capacitance, number of spikes in response to 500 ms current step at double rheobase current, first-spike latency, first to second AP adaptation ratio, and first to last AP adaptation ratio. Silhouette coefficients were calculated using Nbclust R package. Signal analysis scripts available at Marcos Tiago, pax6_codes (2019–2021) online GitHub repository (https://github.com/TiagoMarcos).

## In utero injection and electroporation

Pregnant mice were maintained under inhaled isoflurane anaesthesia for the duration of the procedure. The uterine horns were exposed. In some experiments, 1 to 2 μl of Vismodegib (5 mM solution in DMSO; Stratech) or DMSO alone was injected into the lateral ventricle of each embryo's brain with a glass micropipette. In other experiments, plasmids were injected into the lateral ventricle of each embryo's brain at 1 to 4 mg mL$^{-1}$, the embryo in the uterus was placed between tweezer-type electrodes (CUY650; Nepagene), and an electroporator (CUY21E; Nepagene) was used to deliver 6 pulses (30 V, 50 ms each, 950 ms apart). In both cases, the uterine horns were replaced, the abdominal wall was sutured, and animals recovered. Processing was as described above.

Plasmids used in this study were as follows: CAG-FoxG1-IRES-mCherry (kindly provided by Goishi Myioshi, Tokyo Women's Medical University, Japan); Scrambled shRNA control in pGFP-V-RS shRNA Vector (TR30013, Origene); Smoothened shRNA in pGFP-V-RS shRNA Vector (TG510788, Origene).

## Organotypic slice cultures

Pregnant mice were killed by cervical dislocation following isoflurane overdose and embryos were removed. Embryonic brains were embedded in 3.5% (w/v) low melting point agarose (Lonza) in phosphate buffered saline. Coronal sections were cut at a thickness of 300 μm using a vibratome. The slices were transferred onto a polycarbonate culture membrane (Whatman) floating on minimum essential medium (MEM; Gibco) supplemented with fetal bovine serum (FBS) and incubated at 37°C, 5% $CO_2$. After an hour of incubation, the FBS-supplemented medium was replaced with serum-free neurobasal medium (Gibco) supplemented with B27 (1:50 dilution; Gibco) and N2 supplement (1:100 dilution; Gibco).

In some experiments, after an hour of incubation, the ventral telencephalon of one of the hemispheres was removed from the section while the other hemisphere was left intact. In other experiments, after an hour of incubation, pharmacological treatments were carried out. For Bmp4 treatment, Bmp4 (R&D Systems) or vehicle alone was added to the culture medium to achieve a range of concentrations while maintaining the concentration of all other components constant. Cyclopamine treatment was done in one of 2 ways. Either Affi-gel agarose beads (Bio-Rad) were soaked overnight in cyclopamine (4 mg mL$^{-1}$; Toronto Research Chemicals) or vehicle alone and transferred onto the cortex; or cyclopamine (10μM) or vehicle alone were added in solution to the culture medium. After culture at 37°C, 5% $CO_2$, telencephalic slices were processed for qRT-PCR, immunohistochemistry, or in situ hybridization as described above.

## Dissociated cell culture

Pregnant mice were killed by cervical dislocation and embryos were removed. Tissue samples were taken from each embryo for genotyping to identify whether they were *Pax6* cKO or control [185]. GFP+ cortices were dissociated using a papain dissociation kit (Worthington) and trituration using glass pipettes (BrainBits LLC). Cells were cultured on poly-L-ornithine and laminin-coated coverslips in 24-well plates at 150,000 cells per well in serum-free 2i Medium (Merck). After 24 h, SAG (Abcam) dissolved in DMSO was added to make final SAG concentrations of 1.25 nM, 2.5 nM, 5 nM, 10 nM, 20 nM, 40 nM, 80 nM, or 160 nM. An equivalent number of cultures of each genotype had DMSO only added to give each of the concentrations present at each of the SAG concentrations (0.025%, 0.05%, 0.1%, 0.2%, 0.4%, 0.8%, 1.6%, and 3.2%). Some cultures had both 160 nm SAG and 10 mM EdU added. Cells were fixed with 4% PFA 48 h later.

Antibodies for immunocytochemistry included anti-Tubb3 (1:500; chicken; Novus Biologicals); the others were as described above. DIG-labeled *Dlx1* probes were synthesized and used for fluorescence in situ hybridization as described above. EdU was detected using a Click-iT EdU Alexa Fluor 647 Imaging Kit (Thermo Fisher Scientific). Cells were counterstained with DAPI (1:1,000) and coverslips were mounted with Vectashield HardSet (Vector Laboratories).

All cultures were imaged at ×20 and imported to Fiji software (https://imagej.net/software/fiji/) for cell counting. Each coverslip was divided into 25 areas (S17 Fig) and counting was done using the Cell Counter plug-in in 5 areas selected at random using a random number generator.

## Microscopy

Imaging was carried out using Leica brightfield or epifluorescence microscopes or a Nikon A1R confocal microscope. Microscope images were acquired using the Leica Application Suite X (LAS X) or Nikon NIS-Elements software.

## Supporting information

**S1 Fig. Validation of Pax6 removal and RNAseq data. (A)** Frequently used alleles: *Emx1--Cre$^{ERT2}$* producing TAM-inducible Cre recombinase [33]; *Pax6$^{loxP}$*, from which paired domain-encoding exons were removed by Cre recombinase (*Pax6$^{ΔloxP}$*), rendering it nonfunctional [185]; *RCE$^{EGFP}$*, a Cre reporter producing R26R CAG-boosted EGFP [37]. Mice with deletions in both copies of *Pax6* were designated conditional knockouts (*Pax6* cKOs); those with a deletion in just one copy served as controls. **(B)** qRT-PCR measurements of *Pax6* mRNA levels, normalized to those of *Gapdh*, after TAM administration at E9.5 were used to calculate average (±SEM) ratios between levels in *Pax6* cKO and control littermates (*n* = 3 embryos from 3 litters at each age) (S1 Data). **(C)** Western blots showing Pax6 protein expression in the rostral (R) and caudal (C) cortex of control and *Pax6* cKO littermates at E12.5 and E13.5 after TAM administration at E9.5. **(D)** Quantification of western blots at E12.5 and E13.5 after TAM administration at E9.5. Pax6 protein levels were measured relative to β-actin levels. Average levels (±SEM) were calculated (*n* = 3 independent repeats in each region at each age; in each case, levels in *Pax6* cKOs and controls differed with *p* < 0.01 in Student *t* tests). (Note that Pax6 protein levels in control rostral cortex were almost double those in caudal control cortex at each age, in agreement with previous observations [34] (S1 Data). **(E)** Expression of GFP and Pax6 protein in coronal sections through the cortex of control and *Pax6* cKO embryos at E12.5 and E13.5 after TAM administration at E9.5. GFP was activated by most cortical cells and Pax6 protein was lost from most cortical RGPs across almost the entire cortex, excluding a narrow ventral pallial domain where *Emx1-Cre$^{ERT2}$* was not expressed. Scale bar: 0.1 mm. **(F)** Expression of Pax6 protein in sagittal sections through the cortex of control and *Pax6* cKO embryos at E12.5 and E13.5 after TAM administration at E9.5. Scale bar: 0.1 mm. **(G)** PCA on RNAseq data from CC and CR and *Pax6* cKO caudal (KC) and *Pax6* cKO rostral (KR) cortex at E12.5 and E13.5 showing major clustering based on region and age (raw data are available at the European Nucleotide Archive accession numbers PRJEB5857 and PRJEB6774). **(H)** Significant LFCs with values ≥1 or ≤−1 obtained from bulk RNAseq on samples of E13.5 rostral cortex plotted against significant LFCs calculated from scRNAseq data (by carrying out DEA on average gene expression levels obtained from random subsets of rostral E13.5 *Pax6* cKO and control cells). By way of example, genes are named for some datapoints (raw data are available at the European Nucleotide Archive accession numbers PRJEB27937, PRJEB32740, PRJEB5857 and PRJEB6774). CC, control caudal; CR, control rostral; DEA, differential expression analysis; EGFP, enhanced green fluorescence protein; GFP, green fluorescent protein; LFC, log$_2$ fold change; *Pax6* cKO, *Pax6* conditional knockout; PCA, principal component analysis; qRT-PCR, quantitative real-time polymerase chain reaction; RGP, radial glial progenitor; TAM, tamoxifen.
(TIF)

**S2 Fig. Pax6 loss altered gene expression in cortical cells. (A)** The experimental procedure: TAM was administered at E9.5; one hemisphere from each E12.5 or E13.5 embryo was used to assess Pax6 mRNA and protein levels; the rostral and caudal halves of the other hemisphere were processed for RNAseq. **(B)** Numbers of genes with significantly up-regulated or down-regulated expression levels in caudal (C) and rostral (R) *Pax6* cKO cortex (LFC). **(C)** Numbers of significantly up-regulated or down-regulated genes with a nearby Pax6 BS (raw RNAseq data are available at the European Nucleotide Archive accession numbers PRJEB5857 and PRJEB6774; chromatin immunoprecipitation-sequencing data are from [38]). **(D)** Changes with age in the numbers of significantly up-regulated or down-regulated genes that showed the largest changes in expression levels. We applied a commonly used threshold to include all functionally annotated genes (http://www.ensembl.org/index.html) that at least doubled or

halved their expression levels, i.e., with an LFC in expression $\geq 1$ (for up-regulated genes) or $\leq -1$ (for down-regulated genes), in at least one of the 4 combinations of age and region. This produced a subset of 183 genes: 98 genes were affected at E12.5, 95 remained so, and a further 85 were added at E13.5. To gain an initial impression of biological processes strongly associated with these genes, we passed them through the Database for Annotation, Visualization and Integrated Discovery v6.8 (DAVID v6.8; [66,67]) to obtain sets of significantly enriched GO terms (S2 Table). Some of the GO terms obtained using the up-regulated gene set described the development of cell types normally generated within the telencephalon but outside the cortex in subpallium (where cerebral cortical GABAergic interneurons and, at these ages, oligodendrocytes are made; [9,33,90]). Others described the development of nontelencephalic cell types (spinal cord, inner ear, skeletal system, and neural crest). **(E)** Lists of the 183 genes up-regulated and down-regulated genes that had LFCs $\geq 1$ or $\leq -1$ at E12.5 and/ or E13.5 in at least one of the 4 combinations of age and region (see D above) arranged according to (i) whether the first sign of their up-regulation or down-regulation (which was defined in this case as an adjusted $p < 0.05$, irrespective of LFC magnitude) was at E12.5 or E13.5; and (ii) whether they had a nearby Pax6 BS [38]. We excluded down-regulated genes whose expression levels were very low, based on (i) our E12.5 and E13.5 control cortical RNAseq data; and (ii) expression in sections of cortex from normal embryos of similar ages in the following databases: https://gp3.mpg.de/; https://developingmouse.brain-map.org/; http://www.informatics. jax.org/gxd. We included all up-regulated genes for which there was sufficient documented evidence to allow us to reach valid conclusions on their normal sites of expression at these ages (S3 Table contains citations). Up-regulated genes were separated into those normally expressed by telencephalic cells (Tel) and those normally expressed only outside the telencephalon (Extra-tel). Up-regulated genes shown in boxes were clearly expressed in control cortex, i.e., their cpm were >10: Most up-regulated genes could be considered to be ectopically activated since they had little or no expression in our control RNAseq data from E12.5 and E13.5 cortex. Genes encoding transcription factors were listed using red font. BS, binding site; cpm, counts per million; GO, gene ontology; LFC, $\log_2$ fold change; TAM, tamoxifen. (TIF)

**S3 Fig. Aberrant cell types and ectopic gene expression in E13.5 *Pax6* cKO cortex.** Raw data for (A-F) are available at the European Nucleotide Archive accession number PRJEB32740. **(A)** Violin plots of selected gene expression in each cell type from control and *Pax6* cKO E13.5 cortex. **(B)** Proportions of cells of each type in control and *Pax6* cKO E13.5 cortex. **(C)** The proportions of RGPs and aRGPs in different cell cycle phases in control and *Pax6* cKO E13.5 cortex. We identified each RGP's and aRGP's cell cycle phase by profiling its expression of known cell cycle phase-selective markers [194]. We found that a relatively higher proportion of aRGPs than RGPs were in S phase. This is compatible with previous findings that Pax6 removal causes a shortening of G1, G2, and M phases [35,195]. **(D, E)** Violin plots of the expression levels of genes that were significantly DE between RGPs and aRGPs in different cell cycle phases in control and *Pax6* cKO E13.5 cortex. The expression levels of some of the genes whose expression levels differed between aRGPs and RGPs showed large systematic variation with cell cycle phase; examples are shown in D. The aRGPs' elevated expression of genes such as *Pclaf*, *Rrm2*, *Lig1*, and *Pcna*, whose expression levels increased in S phase, and lowered expression of genes such as *Ube2c*, whose expression levels increased in G2 and M phases, probably reflected the relative increase in the proportions of aRGPs in S phase. However, it was hard to explain all differences between aRGPs and RGPs in this way. For example, levels of *Fos* and *Meg3*, whose expression levels were elevated in aRGPs, and *Neurog2* and *Hes5*, whose expression levels were lowered in aRGPs, showed much less variation with cell cycle phase (E).

**(F)** UMAP plots of the scRNAseq data from *Pax6* cKO and control cells at E13.5, reproduced from Fig 1B, showing cell types and $\log_{10}$-normalized expression of *Fos*. **(G)** Immunohistochemistry for Fos expression in coronal sections of rostral E13.5 lateral and medial control and *Pax6* cKO cortex. Scale bar: 0.05 mm. aRGP, atypical RGP; CRC, Cajal–Retzius cell; DE, differentially expressed; DLN, deep layer neuron; IP, intermediate progenitor; *Pax6* cKO, *Pax6* conditional knockout; RGP, radial glial progenitor; UMAP, uniform manifold approximation and projection. (TIF)

**S4 Fig. Expression profiles of major cell types in *Pax6* cKO and control cortex at E13.5.** Heat map of gene expression in RGPs, aRGPs, IPs, DLNs, and CRCs in E13.5 control and *Pax6* cKO cortex. Raw data are available at the European Nucleotide Archive accession number PRJEB32740. aRGP, atypical RGP; CRC, Cajal–Retzius cell; DLN, deep layer neuron; IP, intermediate progenitor; *Pax6* cKO, *Pax6* conditional knockout; RGP, radial glial progenitor. (TIF)

**S5 Fig. Aberrant cell types and ectopic gene expression in E14.5 *Pax6* cKO cortex.** Raw data are available at the European Nucleotide Archive accession number PRJEB27937. **(A)** Violin plots of selected gene expression in each cell type from control and *Pax6* cKO E14.5 cortex. **(B)** Proportions of cells of each type in control and *Pax6* cKO E14.5 cortex. **(C)** Graphs showing coexpression of markers of normal cortical cells (*Neurog2* and *Eomes*) and normal subcortical cells (*Gsx2* and *Dlx1*) in some aPs in E14.5 *Pax6* cKOs. aP, atypical progenitor; CRC, Cajal–Retzius cell; DLN-L5 and DLN-L6, layer 5 or layer 6 deep layer neurons; DM-IP and DM-SLN, intermediate progenitor or superficial layer neuron in dorsomedial cortex; eGC-P and eCG-N, proliferating or nonproliferating ectopic GABAergic cells; IP, intermediate progenitor; *Pax6* cKO, *Pax6* conditional knockout; RGP, radial glial progenitor; SLN-L2/3 and SLN-L4, layer 2/3 or layer 4 superficial layer neurons. (TIF)

**S6 Fig. Expression profiles of major cell types in *Pax6* cKO and control cortex at E14.5.** Heat map of gene expression in RGPs, aPs, eGC-Ps and eCG-Ns, IPs, DM-IPs and DM-SLNs, SLN-L2/3s and SLN-L4s, DLN-L5s and DLN-L6s, and CRCs in E14.5 control and *Pax6* cKO cortex. Note the increased levels of *Pax6* mRNA in *Pax6* cKO cells, which was anticipated because the loss of Pax6 protein would have removed the negative feedback constraining the transcription of residual *Pax6* coding sequence (Manuel and colleagues, 2007). Raw data are available at the European Nucleotide Archive accession number PRJEB27937. aP, atypical progenitor; CRC, Cajal–Retzius cell; DLN-L5 and DLN-L6, layer 5 or layer 6 deep layer neurons; DM-IP and DM-SLN, intermediate progenitor or superficial layer neuron in dorsomedial cortex; eGC-P and eCG-N, proliferating or nonproliferating ectopic GABAergic cells; IP, intermediate progenitor; *Pax6* cKO, *Pax6* conditional knockout; RGP, radial glial progenitor; SLN-L2/3 and SLN-L4, layer 2/3 or layer 4 superficial layer neurons. (TIF)

**S7 Fig. Integrated scRNAseq data from E14.5 *Pax6* cKO cortex and normal GEs.** UMAP plots of scRNAseq data from normal E14.5 ventral telencephalon [86] integrated with our scRNAseq data from E14.5 *Pax6* cKO cortex (raw data are available at the European Nucleotide Archive accession number PRJEB27937). Abbreviations for cortex in S5A Fig. CGE, caudal ganglionic eminence; GE, ganglionic eminence; LGE, lateral ganglionic eminence; MGE, medial ganglionic eminence; *Pax6* cKO, *Pax6* conditional knockout; SVZ, subventricular zone; UMAP, uniform manifold approximation and projection; VZ, ventricular zone. (TIF)

**S8 Fig.** *Ascl1*, *Neurog2*, **and** *Prdm13* **expression in control and** *Pax6* **cKO cortex. (A)** In situ hybridizations for *Ascl1* and *Neurog2* in control and *Pax6* cKO cortex at E13.5 and E14.5. Scale bars: 0.1 mm. **(B)** Method for obtaining a surface-view reconstruction of labeling across the cortex from a series of equally spaced coronal sections (s1 to s(n)). Relative intensity of label (I) or labeled cell number (N) were measured in areas (a1 to a(n)) with average width (through cortical depth) of 0.1 mm in each section. Values were then laid out on a flattened representation of the cortical surface and maps generated by interpolation. **(C)** Quantification showing the similarity in the distributions with depth of the proportions of cells expressing *Neurog2* mRNA in control lateral cortex and *Ascl1* mRNA in *Pax6* cKO lateral cortex at E13.5 and E14.5. Data were obtained using 25 μm × 100 μm bins, as shown, from the lateral cortex in centrally located coronal sections through the brains of 4 *Pax6* cKOs and 4 controls at E13.5 and 3 *Pax6* cKOs and 3 controls at E13.5. Data points are for individual animals and shaded areas show the range of values with depth (S2 Data). **(D)** In situ hybridization for *Prdm13* in rostral and caudal *Pax6* cKO cortex at E16.5 after tamoxifen administration at E9.5. Scale bar: 0.1 mm. *Pax6* cKO, *Pax6* conditional knockout. (TIF)

**S9 Fig. Pax6 loss induced ectopic gene expression in distinct spatiotemporal patterns. (A)** Immunoreactivity for Gsx2 in control and *Pax6* cKO cortex between E12.5 and E16.5 following tamoxifen at E8.5, E9.5, E10.5, or E13.5 and in the cortex of E13.5 constitutive *Pax6*$^{-/-}$ embryos. Scale bar: 0.1 mm. **(B, C)** In situ hybridizations for *Dlx1* and *Gad1* in control and *Pax6* cKO cortex at E12.5 and E14.5 following tamoxifen at E9.5. Scale bars: 0.1 mm. **(D)** In situ hybridizations for *Gad1* and immunoreactivity for GFP in control and *Pax6* cKO cortex at E14.5 following tamoxifen at E9.5, showing enlargement of boxed area with and without *Gad1* staining. Red arrows in *Pax6* cKO cortex: examples of *Gad1*+ cells that were GFP-negative. Scale bars: 0.1 mm and 0.01 mm. **(E)** Fluorescence and colorimetric immunoreactivity for Ascl1 and Gsx2 and in situ hybridizations for *Dlx1* and *Gad1* in *Pax6* cKO lateral cortex at E14.5 following tamoxifen at E9.5. Scale bars: 0.1 mm. **(F)** Summary of gene expression changes in (F) with time following Pax6 deletion. **(G)** Fluorescence and colorimetric immunoreactivity for Gsx2 and Eomes and in situ hybridizations for *Dlx1* and *Gad1* in *Pax6* cKO cortex at E13.5 and E14.5 following tamoxifen at E9.5. Scale bar: 0.01 mm. **(H, I)** Colorimetric and fluorescence immunoreactivity for Olig2 and Ascl1 in control and *Pax6* cKO cortex between E13.5 and E16.5 following tamoxifen at E9.5, E10.5 or E13.5. Scale bars: 0.1 mm. GFP, green fluorescent protein; *Pax6* cKO, *Pax6* conditional knockout. (TIF)

**S10 Fig. Pax6 deletion and Olig2 expression in** *Pax6* **cKO cortex. (A)** Expression of Pax6 protein in coronal sections through the cortex of control and *Pax6* cKO embryos at E13.5 after tamoxifen administration at E10.5 and at E16.5 after tamoxifen administration at E13.5. Scale bar: 0.1 mm. **(B)** Expression of Pax6 protein and GFP in sections through the *Pax6* cKO cortex at E16.5 after tamoxifen administration at E13.5. The vast majority of cortical RGPs were Pax6-negative. Many of the few remaining Pax6+ cells were GFP-negative (examples marked with white dots). Scale bar: 0.01 mm. **(C)** UMAP plots from *Pax6* cKO cortex at E14.5 showing relative expression and coexpression levels of *Olig2*, *Gsx2*, *Dlx1*, and *Gad1* in each cell (on a 0–10 scale, 10 was the highest expression level of the gene in question). Raw data are available at the European Nucleotide Archive accession number PRJEB27937. GFP, green fluorescent protein; *Pax6* cKO, *Pax6* conditional knockout; RGP, radial glial progenitor; UMAP, uniform manifold approximation and projection. (TIF)

**S11 Fig. Analyses of eGC fates. (A)** Colorimetric immunoreactivity for Gsx2 and in situ hybridization for *Gad1* in E14.5 *Pax6* cKO lateral cortex after tamoxifen[E9.5]. Scale bar: 0.1 mm. **(B)** Fluorescence immunoreactivity for Gsx2 and GFP, the latter marking *Gsx2*-lineage cells, in E14.5 control and *Pax6*[−/−] telencephalon. Scale bar: 0.1 mm. **(C, D)** Fluorescence immuno-reactivity for GFP+ (*Gsx2*-lineage) and in situ hybridization for *Gad1+* cells in E16.5 control and *Pax6*[−/−] lateral cortex. Scale bars: 0.1 mm and 0.01 mm. **(E)** Three *Gad1+* cells in the CP of an E18.5 *Pax6* cKO (tamoxifen[E9.5]) carrying the *RCE*[EGFP] Cre-reporter. One was *Emx1*-line-age (GFP+). Scale bar: 0.01 mm. **(F)** Experimental design for analysis of the effects of Pax6 deletion on cortical cell numbers (Fig 4H). The *Emx1-Cre*[ERT2] allele with tamoxifen[E9.5] was used to delete *Pax6*; embryos carried a GFP reporter of Cre activity; embryos were collected at E18.5; regularly spaced coronal sections were double-stained for GFP protein and *Gad1* mRNA. We measured the surface area of the CP of the lateral cortex and the surface area of the sub-CP masses of one hemisphere in each section and interpolated to estimate the volumes of these structures in each embryo. We used random sampling (e.g., white boxes) to measure the average densities of cells in each region. We then calculated their total numbers in each region in each embryo. Data from 4 littermate pairs from separate mothers were used to produce Fig 4H. **(G)** Fluorescence immunoreactivity for GFP (*Emx1*-lineage) and in situ hybrid-ization for *Gad1+* cells in P34 control and *Pax6* cKO cortex after tamoxifen at E9.5. High-magnification images: same region of CP with and without *Gad1* staining. Scale bar: 1 mm and 0.1 mm. **(H)** Immunoreactivity for Caspase-3 in CP and in sub-CP masses in *Pax6* cKO cortex at E16.5 and at P10 after tamoxifen at E9.5. Scale bar: 0.1 mm. CP, cortical plate; eGC, ectopic GABAergic cell; GFP, green fluorescent protein; *Pax6* cKO, *Pax6* conditional knock-out.
(TIF)

**S12 Fig. Comparing the effects of losing Pax6 alone versus Pax6 and ectopic Gsx2. (A)** Alleles used to delete conditionally *Gsx2*: *Emx1-Cre*[ERT2] producing TAM-inducible Cre recombinase [33]; *Gsx2*[loxP] [47]; *RCE*[EGFP] [37]. **(B, C)** Immunoreactivity for Ascl1 and in situ hybridization for *Neurog2* in *Pax6* cKO and *Pax6 Gsx2* dcKO cortex at E13.5 and E14.5. Scale bar: 0.1 mm. **(D)** In situ hybridization for *Gsx1* in control, *Pax6* cKO and *Pax6 Gsx2* dcKO cortex at E13.5. Scale bar: 0.1 mm. **(E)** PCA on RNAseq data from *Pax6* cKO (*n* = 5 embryos) and *Pax6 Gsx2* dcKO (*n* = 5 embryos) cortex at E13.5. Data were from the rostral half of the cortex, where the proportion of cells activating *Gsx2* after *Pax6* deletion was highest. Raw data are available at the European Nucleotide Archive accession number PRJEB21105. **(F)** A list of all genes that were significantly down-regulated or up-regulated (adjusted *p* < 0.05) in *Pax6 Gsx2* dcKO cortex compared to *Pax6* cKO cortex and their LFCs in *Pax6* cKO compared to control cortex (n.s., not significant). Raw data are available at the European Nucleotide Archive accession numbers PRJEB21105 and PRJEB5857. dcKO, double conditional KO; LFC, log$_2$ fold change; *Pax6* cKO, *Pax6* conditional knockout; PCA, principal component analysis; TAM, tamoxifen.
(TIF)

**S13 Fig. Widespread loss of gene expression in Pax6 cKOs. (A)** TAM was administered at E9.5 to generate control and *Pax6* cKO embryos; brains were sectioned sagittally at E13.5. **(B)** In situ hybridizations for *Pde1b*, *Rlbp1*, *Igsf11*, *Heg1*, and *Neurod4* in E13.5 control and *Pax6* cKO cortex. Scale bar: 0.1 mm. *Pax6* cKO, *Pax6* conditional knockout; TAM, tamoxifen.
(TIF)

**S14 Fig. CP development from *Pax6* cKO RGPs. (A)** The experimental procedure for (B): TAM was administered at E9.5 to generate control and *Pax6* cKO embryos; BrdU was injected

on either E13.5 or E16.5; brains were sectioned coronally and analysed at P10. **(B)** BrdU-labeled cells in P10 lateral CP; experimental procedure in (A). Scale bar: 0.1 mm. **(C)** Immunoreactivity for Slc17a7 (formerly Vglut1) in E18.5 control and *Pax6* cKO cortex after TAM at E9.5. Scale bar: 0.1 mm. **(D)** The experimental procedure for (E-I): slices from P5–13 *Emx1*-Cre; *RCE^EGFP*; *Pax6^loxP/+* (control) or *Pax6^loxP/loxP* (*Pax6* cKO) mice were used for electrophysiology. **(E)** GFP+ cells in layers 2/3 and 5 of somatosensory cortex area 1 (S1) were targeted for whole-cell current clamp recordings (electrodes are visible in inset in upper panel and targeting a GFP+ cell in lower panel). **(F)** Examples of membrane voltage responses to progressive current injections for control and *Pax6*cKO cortex (500 ms square steps; hyperpolarizing step: −25 pA; depolarizing steps: rheobase and double rheobase). **(G)** Unsupervised hierarchical clustering analysis for S1 layer 5. Features used for clustering are listed in S5 Table. Purple tones represent older pups (P8-P10). Green tones represent younger pups (P5-P7). Genotype bar indicates control cells (gray; *n* = 54) and *Pax6*cKO cells (red; *n* = 55). Cells from both genotypes were spread across the clusters with no segregation of *Pax6*cKO cells. Silhouette coefficient analysis suggested the optimal number of clusters was 2 (silhouette coefficient = 0.26, k = 2), which separated cells mainly by age (Sheet C in S5 Data). **(H)** Unsupervised hierarchical clustering analysis for S1 layers 2/3. Features used for clustering are listed in S5 Table. Silhouette coefficient analysis suggested the optimal number of clusters was 2 (silhouette coefficient = 0.44, k = 2). Clustering split the cells into 2 main branches, with one containing 3 cells all from *Pax6*cKO mice, but this separation might have occurred by chance (Barnard unconditional two-tailed test, *p* = 0.09) (Sheet C in S5 Data). **(I)** S1 layer 2/3 GFP+ fast spiking cell; double rheobase current injection response. BrdU, 5-bromo-2′-deoxyuridine; CP, cortical plate; *Pax6* cKO, *Pax6* conditional knockout; RGP, radial glial progenitor; TAM, tamoxifen. (TIF)

**S15 Fig. Expression and effects of Shh on eGC production. (A)** Flattened surface views of the cortex showing the densities of *Gad1*+ cells at E13.5 and E14.5 in controls: *Gad1*+ cells spread in increasing numbers across the cortex from lateral to medial with a similar spatiotemporal pattern to the spread of *Gsx2* and *Dlx1* activation after TAM at E9.5 (maps for Gsx2 and *Dlx1* reproduced from Fig 3H). **(B)** Validation of the Shh antibody: comparison of staining patterns, in both neural and nonneural tissues [196], with in situ hybridization patterns from the Allen Brain Atlas in E11.5 and E13.5 controls and lack of staining in E13.5 $Shh^{-/-}$ mutants (kindly provided by Laura Lettice and Bob Hill, Edinburgh University). Scale bar: 0.1 mm. **(C)** Immunoreactivity for Shh in control telencephalon at E14.5 and in *Pax6* cKO telencephalon after TAM at E9.5 at E14.5 and after 2 DIV from E13.5 (Fig 6A). Scale bars: 0.1 mm and 0.01 mm. **(D)** Following the experimental procedure reproduced from Fig 6K: constructs expressing *Smo* shRNA + GFP or scrambled shRNA + GFP were electroporated into the cortex of E14.5 *Pax6* cKO embryos made using *Emx1-Cre*; electroporated cells were analysed at E15.5. Blind to Gsx2 expression, we identified 80–100 GFP+ cells and a random selection of 80–100 intermingled GFP− cells in each of 3 embryos from 3 litters given *Smo* shRNA and each of 4 embryos from 3 litters given scrambled shRNA. The intensity of Gsx2 immunoreactivity was then measured in all of these cells and frequency distributions of intensities in electroporated versus non-electroporated cells were compared in brains that received *Smo* shRNA and in brains that received scrambled shRNA (results in Fig 6M). DIV, day in vitro; eGC, ectopic GABAergic cell; GFP, green fluorescent protein; *Pax6* cKO, *Pax6* conditional knockout; TAM, tamoxifen. (TIF)

**S16 Fig. *Foxg1* deletion and codeletion with *Pax6*. (A)** Deletion of *Foxg1*, here with *Emx1--Cre^ERT2* and TAM, removes its coding region [184] and activates the Cre reporter. **(B)** TAM

was administered at E9.5 to generate *Pax6* cKOs in which both, one, or neither *Foxg1* allele(s) were also deleted; brains were analysed at E13.5 and E14.5. The *Pax6^loxP^* allele was shown in S1A Fig. **(C)** In situ hybridizations for *Foxg1* and immunohistochemistry for Foxg1 and Pax6 following deletion of both, one or neither *Foxg1* allele(s) at E13.5 and E14.5. Arrows: A few cells remained undeleted and formed small clones expressing both Foxg1 and Pax6. Scale bars: 0.1 mm. **(D)** Immunoreactivity for Ascl1 in E14.5 cortex from *Pax6* cKO embryos in which neither, one or both *Foxg1* allele(s) were deleted by TAM at E9.5. Scale bars: 0.1 mm. **(E)** Immunoreactivity for Eomes in E16.5 lateral cortex from control embryos and *Pax6 Foxg1* double KOs. Scale bar: 0.1 mm. **(F)** Proportions of cells in the ventricular and subventricular zones of E16.5 lateral cortex expressing Eomes in control embryos and embryos of the 3 genotypes in (B) (averages ± SD of $n = 3$; *Pax6* single cKO average was significantly lower than all others, $p < 0.05$ in all comparisons; Student $t$ tests) (Sheet D in S7 Data). **(G)** Results of experiment in Fig 7F: coexpression of Eomes and mCherry in a coronal section. Scale bar: 0.1 mm. *Pax6* cKO, *Pax6* conditional knockout; TAM, tamoxifen.
(TIF)

**S17 Fig. Quantification method for dissociated cultures.** Method for quantification of the effects of SAG on numbers of GFP+ cells expressing various markers: Cells were cultured on coverslips and, after fixation and reaction, counting grids were used to sample 5 randomly selected areas from each coverslip. Several independent biological repeats were used for each condition (i.e., each concentration of SAG or vehicle alone, on *Pax6* cKO or control cells) (Fig 8). GFP, green fluorescent protein; *Pax6* cKO, *Pax6* conditional knockout; SAG, signaling agonist.
(TIF)

**S18 Fig. Phospho-Smad1/5/9 expression and Bmp4 effects on Olig2 expression in control and *Pax6* cKO cortex. (A)** Immunoreactivity for phospho-Smad1/5/9 in E14.5 control and *Pax6* cKO cortex after tamoxifen^E9.5^. Scale bar: 0.1 mm. **(B)** Concentration-response measured using qRT-PCR: *Olig2* levels (averages ± SEM; values are relative to the average level in control cortex treated with 0 Bmp4) in control and *Pax6* cKO slices with increasing concentrations of Bmp4 ($n = 3$ independent cultures at each concentration). Two-way ANOVA showed significant effects of genotype on *Olig2* ($p < 0.005$), but no significant effect of Bmp4 concentration and no significant interaction effect. Differences between genotypes at each Bmp4 concentration were tested with Bonferroni's method for comparison of means (*, $p < 0.05$; **, $p < 0.01$; ***, $p < 0.005$) (S9 Data). *Pax6* cKO, *Pax6* conditional knockout; qRT-PCR, quantitative real-time polymerase chain reaction.
(TIF)

**S1 Table. Results of bulk RNAseq: log$_2$ fold changes of genes that were significantly differentially expressed between control and *Pax6* cKO cortex (adjusted p, or padj, <0.05) rostrally and caudally at E12.5 and E13.5.**
(XLSX)

**S2 Table. Significantly enriched gene ontology (GO) terms obtained by passing through DAVID v6.8 all functionally annotated genes with a LFC in expression ≥1 (for up-regulated genes) or ≤−1 (for down-regulated genes) in at least one of the 4 combinations of age and region studied.**
(XLSX)

**S3 Table. Citations providing evidence on the normal sites of expression of genes that were up-regulated in *Pax6* cKO RNAseq datasets.**
(DOCX)

**S4 Table. Genes whose expression levels were the most different between RGPs and aRGPs (average LFCs >0.3 or <−0.3) and GO terms obtained by passing this list through DAVID.**
(XLSX)

**S5 Table. Intrinsic properties of primary somatosensory cortex (S1) layer 5 cells at P5–7 and P8–P10 and layer 2/3 cells at P10–13.** Values are means ± 95% confidence intervals.
(XLSX)

**S1 Data. Data referred to in S1 Fig.**
(XLSX)

**S2 Data. Data referred to in S8 Fig.**
(XLSX)

**S3 Data. Data referred to in text.**
(XLSX)

**S4 Data. Data referred to in Fig 4.**
(XLSX)

**S5 Data. Data referred to in Figs 4 and S14.**
(XLSX)

**S6 Data. Data referred to in Fig 6.**
(XLSX)

**S7 Data. Data referred to in Figs 6, 7, and S16 and text.**
(XLSX)

**S8 Data. Data referred to in Fig 8.**
(XLSX)

**S9 Data. Data referred to in Figs 9 and S18.**
(XLSX)

**S1 Raw Images. Western blots shown in S1 Fig.**
(TIF)

## Acknowledgments

We thank Ugo Borello, Matt Colligan, Petrina Georgala, Anisha Kubasik-Thayil, Da Mi, Soham Mitra, Elena Purlyte, Thomas Theil, Dafni Triantafyllou, and staff in Bioresearch and Veterinary Services and Edinburgh Genomics for their contributions.

## Author Contributions

**Conceptualization:** Kai Boon Tan, David J. Price.

**Data curation:** Martine Manuel, Kai Boon Tan, Zrinko Kozic, Michael Molinek, Maizatul Fazilah Abd Razak, Dániel Dobolyi, David J. Price.

**Formal analysis:** Martine Manuel, Kai Boon Tan, Zrinko Kozic, Tiago Sena Marcos, Maizatul Fazilah Abd Razak, Dániel Dobolyi.

**Funding acquisition:** David J. Price.

**Investigation:** Martine Manuel, Kai Boon Tan, Tiago Sena Marcos, Maizatul Fazilah Abd Razak, Dániel Dobolyi, Ross Dobie, Beth E. P. Henderson, Neil C. Henderson, Wai Kit Chan, Michael I. Daw, John O. Mason.

**Methodology:** Martine Manuel, Kai Boon Tan, Zrinko Kozic, Michael Molinek, Tiago Sena Marcos, Maizatul Fazilah Abd Razak, Dániel Dobolyi, David J. Price.

**Project administration:** Martine Manuel, David J. Price.

**Resources:** Michael Molinek, David J. Price.

**Software:** Zrinko Kozic, Tiago Sena Marcos.

**Supervision:** Martine Manuel, Kai Boon Tan, Zrinko Kozic, Michael Molinek, Wai Kit Chan, Michael I. Daw, John O. Mason, David J. Price.

**Writing – original draft:** David J. Price.

**Writing – review & editing:** David J. Price.

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
