## [Editor Report · Decision Letter 0]

1 Feb 2022

Dear Dr Price, 

Thank you for submitting your manuscript entitled "Pax6 limits the competence of developing cerebral cortical cells" for consideration as a Research Article by PLOS Biology.

Your manuscript has now been evaluated by the PLOS Biology editorial staff as well as by an academic editor with relevant expertise and I am writing to let you know that we would like to send your submission out for external peer review.

Once your full submission is complete, your paper will undergo a series of checks in preparation for peer review. Once your manuscript has passed the checks it will be sent out for review. To provide the metadata for your submission, please Login to Editorial Manager (https://www.editorialmanager.com/pbiology) within two working days, i.e. by Feb 03 2022 11:59PM.

If your manuscript has been previously reviewed at another journal, PLOS Biology is willing to work with those reviews in order to avoid re-starting the process. Submission of the previous reviews is entirely optional and our ability to use them effectively will depend on the willingness of the previous journal to confirm the content of the reports and share the reviewer identities. Please note that we reserve the right to invite additional reviewers if we consider that additional/independent reviewers are needed, although we aim to avoid this as far as possible. In our experience, working with previous reviews does save time. 

If you would like to send previous reviewer reports to us, please email me at ialvarez-garcia@plos.org to let me know, including the name of the previous journal and the manuscript ID the study was given, as well as attaching a point-by-point response to reviewers that details how you have or plan to address the reviewers' concerns. 

Given the disruptions resulting from the ongoing COVID-19 pandemic, please expect some delays in the editorial process. We apologise in advance for any inconvenience caused and will do our best to minimize impact as far as possible.

Kind regards,

Ines

--

Ines Alvarez-Garcia, PhD

Senior Editor

PLOS Biology

---

## [Decision Letter · Decision Letter 1]

28 Mar 2022

Dear Dr Price,

Thank you for submitting your manuscript entitled "PAX6 limits the competence of developing cerebral cortical cells" for consideration as a Research Article at PLOS Biology. Thank you also for your patience as we completed our editorial process, and please accept my apologies for the delay in providing you with our decision. Your manuscript has been evaluated by the PLOS Biology editors, an Academic Editor with relevant expertise, and by two independent reviewers.

As you will see, the reviewers find the conclusions novel and interesting, but they also ask for several clarifications, some of them requiring additional data. In addition, the reviewers think you should streamline the text to make it accessible for a broad audience.

In light of the reviews (attached below), we are pleased to offer you the opportunity to address the comments from the reviewers in a revised version that we anticipate should not take you very long. We will then assess your revised manuscript and your response to the reviewers' comments and we may consult the reviewers again.

We expect to receive your revised manuscript within 1 month.

**IMPORTANT - SUBMITTING YOUR REVISION**

3. Resubmission Checklist

a) *PLOS Data Policy*

b) *Published Peer Review*

Sincerely,

Ines

--

Ines Alvarez-Garcia, PhD

Senior Editor

PLOS Biology

Reviewers' comments

Rev. 1:

The manuscript by Manuel et al. titled "Pax6 limits the competence of developing cerebral cortical cells" systematically dissects the role of Pax6 in corticogenesis by using a conditional knockout mouse model. The authors conclude that Pax6 expression is not necessary for the formation of excitatory neurons but rather acts as a protective factor to block extrinsic signals leading to altered cell fates. This work is quite extensive and reaffirms many results from prior studies thus impacting the overall novelty. However, with addition of the single cell data, this work can be an important modern resource for those studying corticogenesis.

Comments (mainly for clarity):

1. In the transcriptomic analysis, it is unclear the decision to choosing the thresholds for the log fold changes used to call differential gene expression. For example, the LFC > 0 in Figure 1B vs LFC >= 1 in Figure 1E.

2. It is unclear in the text/methods how the aRGP population was determined as a different cell type from RGPs in the UMAP (Figure 2B). It may be helpful to more broadly define the characteristics of aRGPs compared to RGPs.

3. In Figure 4, the RNA velocity analysis shows that RGPs and IP cells move toward deep and superficial layer neuron cell types, but it is a bit confusing that the directionality continues from superficial layer neurons to deep layer identity.

4. In the embryonic data, it would help to validate some of the aRGP cells within the developing cortex, like Fos or Meg3.

5. The interpretation of the Bmp treatment in Pax6 cKO cortices suggest that Bmp4 can suppress the ventral marker Gsx2. Is there any data that can distinguish between suppression versus simply the differentiation of Gsx2+ cells?

Rev. 2:

In this manuscript, Manuel et al. describe the role of Pax6 development of the neocortex. They analyzed transriptome changes in different cell types of Pax6 mutant cortex by single cell RNA Seq as well as by in situ hybridization and immunohistochemistry.

They also carried out in vitro experiments aiming to test responsiveness of Pax6 deficient cells to developmental morphogenes, such as Shh and BMP.

The main conclusion of the manuscript is that Pax6 controls competence of cortical cells to environmental cues.

Essentially the manuscript can be divided into two parts: the first part describes Pax6 mutant phenotype in the developing neocortex, the second part investigates the role of in vitro and in vivo manipulations of necortical tissue and cells in the presence of activators/inhibitors of SHH and BMP signaling pathways.

In the first part , Figures 1-9 present very comprehensive and thorough analysis of molecular changes at single cell type level in the Pax6 mutant neocortex. This analysis is followed by spatial expression analysis of some selected genes. I should admit that this part does not contain conceptually novel data. Pax6 role in the neocortex is very well studied and most observations presented in this part have been reported in numerous studies mentioned by the authors. This part is lengthy and very difficult to comprehend for non-experts in the field of neocortical development. I think that this part can be substantially reduced and half of the figures can go to the supplement. I think the authors should make this part more focused on eGCs cells in Pax6 mutant. Another aspect that is important for the main message of the manuscript, is presence of glutamatergic neurons in Pax6 mutants. All the data that confirm or repeat published observations, like "ventralization" of the dorsal telencephalon, proliferation characteristics of Pax6 deficient cells etc. do not have to be shown and discussed so extensively. Instead I suggest to focus on the main message of this part that is important to understand the second part of the manuscript.

The second part of the manuscript, where the authors test the hypothesis that Pax6 and Foxg1 modulate responsiveness of neocortical cells to external factors, Shh and BMP is indeed novel. In this part the authors use several methods and convincingly demonstrate that the number of misspecified, interneuron-like Gsx2 positive cells depend on Shh signal. They also provided evidence that the source of Shh is migrating interneurons. On the other hand, they show that Foxg1 in contrast to Pax6 is a positive regulator of Gsx2 fate. Additionally, they demonstrated that BMP plays an opposite to Shh role on Gsx2 fate.

In sum: the manuscript presents very comprehensive study, but substantial part of it even though performed with state-of- the art technology, and provided deeper insight into molecular characterisation of Pax6 mutation is not novel. On the other hand it does provide conceptually novel data on interactions of cell intrinsic and cell extrinsic factors in the developing neocortex.

My recommendation: to shorten the manuscript substantially, with more focus on novel findings and reduce the number of figures to seven-eight.

---

## [Decision Letter · Decision Letter 2]

16 Jun 2022

Dear Dr Price,

Thank you for your patience while we considered your revised manuscript entitled "PAX6 limits the competence of developing cerebral cortical cells" for publication as a Research Article at PLOS Biology. This revised version of your manuscript has been evaluated by the PLOS Biology editors, the Academic Editor and the two original reviewers.

Based on the reviews, we are likely to accept this manuscript for publication, provided you satisfactorily address the data and other policy-related requests stated below.

In addition, we would like you to consider a suggestion to improve the title:

"PAX6 limits the competence of developing cerebral cortical cells to respond to inductive intercellular signals"

We expect to receive your revised manuscript within two weeks. 

*Published Peer Review History*

*Press*

Sincerely,

Ines

--

Ines Alvarez-Garcia, PhD

Senior Editor,

ialvarez-garcia@plos.org,

PLOS Biology

ETHICS STATEMENT:

Thank you for including the ethics statement. Please include also an approval number.

DATA POLICY:

Thank you for sending the data files containing the data underlying the graphs shown in the figures. However, we are missing some of them and the labelling in the data files is very confusing. Please add the missing data and check all the files to make sure the figures are numbered correctly. In addition, please check that all the figure legends accurately describe where the data can be found.

Please add or clarify where the data of these figures can be found:

Fig. S1B, D, G, H; Fig. S2C; Fig. S3A-E; Fig. S4; Fig. S5B, C; Fig. S6; Fig. S8C; Fig. S12C and Fig. S18B

Thank you for sending a file with the gels. We do require the original, uncropped and minimally adjusted images supporting all blot and gel results reported in an article's figures or Supporting Information files. Please carefully read our guidelines for how to prepare and upload this data: https://journals.plos.org/plosbiology/s/figures#loc-blot-and-gel-reporting-requirements

Reviewers' comments

Rev. 1:

The revision by Manuel et al. has addressed and clarified the concerns raised in the first submission. In addition, the authors carried out new experiments toward validating the single cell gene expression changes in aRGP cells.

The article as a whole reads better and the major points in the result come across clearer. While at times, sections such as "Why Pax6 deletion altered the fates of only some cortical cells: a hypothesis" could be made more succinct, I would defer to the editorial group on these decisions. Again, this work will be a good resource for those interested in the role of pax6 in cortical development.

Rev. 2:

The authors tried to change the manuscript according to my suggestions. They moved some figures to the supplement and modified the text.

However, I am still not convinced that they need to show expression changes data that have been published before. Figures showing downregulation of Eomes and Gsx, for example, can go to the supplement.

---

## [Editor Report · Decision Letter 3]

8 Jul 2022

Dear Dr Price,

Thank you for the submission of your revised Research Article entitled "PAX6 limits the competence of developing cerebral cortical cells to respond to inductive intercellular signals" for publication in PLOS Biology. On behalf of my colleagues and the Academic Editor, Bassem Hassan, I am happy to say that we can in principle accept your manuscript for publication, provided you address any remaining formatting and reporting issues. These will be detailed in an email you should receive within 2-3 business days from our colleagues in the journal operations team; no action is required from you until then. Please note that we will not be able to formally accept your manuscript and schedule it for publication until you have completed any requested changes.

PRESS

Sincerely, 

Ines

--

Ines Alvarez-Garcia, PhD, PhD

Senior Editor

PLOS Biology
